# From Uncontextualized Embeddings to Marginal Feature Effects: Incorporating Intelligibility into Tabular Transformer Networks

## Abstract

In recent years, deep neural networks have showcased their predictive power across a variety of tasks. Beyond natural language processing, the transformer architecture has proven efficient in addressing tabular data problems and challenges the previously dominant gradient-based decision trees in these areas. However, this predictive power comes at the cost of intelligibility: Marginal feature effects are almost completely lost in the black-box nature of deep tabular transformer networks. Alternative architectures that use the additivity constraints of classical statistical regression models can maintain intelligible marginal feature effects, but often fall short in predictive power compared to their more complex counterparts. To bridge the gap between intelligibility and performance, we propose an adaptation of tabular transformer networks designed to identify marginal feature effects. We provide theoretical justifications that marginal feature effects can be accurately identified, and our ablation study demonstrates that the proposed model efficiently detects these effects, even amidst complex feature interactions. To demonstrate the model's predictive capabilities, we compare it to several interpretable as well as black-box models and find that it can match black-box performances while maintaining intelligibility. The source code is vailable at `https://anonymous.4open.science/r/nmfrmr-B086`.

## 1 Introduction

Interpretability has emerged as one of the core concepts of tabular data analysis. Especially in high-risk domains such as healthcare, where understanding the data's underlying effects is of crucial importance, this leads researchers to commonly rely on identifiable and interpretable generalized additive models (GAMs) (Hastie, 2017), instead of powerful neural networks or decision trees (Erfanian et al., 2021; Ravindra et al., 2019; Prata et al., 2020). In applications where predictive power is a central objective, researchers often resort to model-agnostic methods that try to explain model predictions via local approximation and feature importance like Locally Interpretable Model Explanations (LIME) (Ribeiro et al., 2016), or Shapley values (Shapley, 1953) and their extensions (Sundararajan & Najmi, 2020). While these methods are very effective for, e.g., image classification tasks, they can be hard to interpret for tabular regression problems.

Although predictive modeling in the domain of tabular data is traditionally dominated by tree-based bagging and boosting approaches (Breiman, 2001; Chen & Guestrin, 2016; Prokhorenkova et al., 2018), several relatively recent results show that deep-learning based techniques can be highly competitive in general or even superior on specific tabular datasets (McElfresh et al., 2024). In particular models utilizing the transformer architecture stand out in terms of their predictive power (Gorishniy et al., 2021; 2023; Hollmann et al., 2022). The most performant models, Tabular (bayesian) prior-fitted transformer models (Hollmann et al., 2022), can only be used for smaller datasets. However, FT-Transformers have robustly proven to be performant on tabular problems (Gorishniy et al., 2021; Grinsztajn et al., 2022; McElfresh et al., 2024). Nevertheless, despite the use of [CLS] tokens, allowing for heuristic interpretability of feature importance, these models remain black boxes and do not provide insights into marginal feature effects.

To bridge the gap in performance seen with traditional statistical models while preserving interpretability, recent efforts have focused on enhancing visual interpretability by incorporating additivity constraints into neural network architectures (Agarwal et al., 2021; Chang et al., 2021; Enouen & Liu, 2022). Similar to GAMs each feature is fit with a separate shape function. Neural additive models (NAMs) (Agarwal et al., 2021) and their extensions have emerged as a powerful yet interpretable solution for tabular data problems. Depending on the model, shape functions vary from Multi-layer Perceptrons (MLPs) (Agarwal et al., 2021; Radenovic et al., 2022; Thielmann et al., 2023) to neural oblivious decision trees (Chang et al., 2021), splines (Luber et al., 2023; Rügamer et al., 2023; 2021; Dubey et al., 2022), ensemble decision trees (Nori et al., 2019) or transformer networks (Thielmann et al., 2024). While these models offer visual interpretability, they come with inherent downsides: I) There is a performance gap relative to fully connected black-box models and even to simple MLPs. II) The networks can become parameter-dense, depending on the complexity of the marginal effects, as each feature is modeled with a distinct shape function (Agarwal et al., 2021; Thielmann et al., 2023; 2024). III) The complexity of the model structure grows rapidly with the number of features, especially when accounting for feature interactions, leading not only to a potentially suboptimal inductive bias, but also a vast hyperparamter space making effective hyperparamter tuning computationally expensive or even impossible. IV) Additionally, higher-order feature interactions can negatively impact the model's identifiability and are thus are often simply excluded (Kim et al., 2022; Siems et al., 2024).

We propose to leverage the existing proven, and highly performant architectures for deep tabular learning and introduce a new architecture to bridge the gap between high-performing tabular models and inherently interpretable models using the flexible tabular deep learning architecture from Gorishniy et al. (2021). More specifically, we use target-aware embeddings (Gorishniy et al., 2022) and fit shallow one layer neural networks on uncontextualized embeddings while accounting for all higher order interaction effects.

Our contributions can be summarized as follows:

I. We introduce the NAMformer, a fully connected tabular deep learning architecture that combines a powerful FT-Transformer with interpretable feature networks.

II. We demonstrate that this straightforward approach yields intelligible and identifiable marginal feature effects, while perfectly maintaining the predictive power of FT-Transformers and adding an almost negligible amount of additional parameters to the model.

III. We show that identifiability can be achieved by employing strategic feature dropout.

## 2 METHODOLOGY

The core of the NAMformer architecture is given by an FT-Transformer in combination with a shallow MLP that both take uncontextualized embeddings as their input. In a nutshell, all numerical features are encoded and all categorical features are tokenized. Subsequently all features are fed through data type dependent embedding layers. The embeddings are passed through a stack of transformer layers, after which the [CLS] token embedding is processed by a task specific model head. The *uncontextualized* embeddings, before being passed through the transformer layers, are fed to shallow, one-layer independent neural networks. The final prediction is gained by summation over all shallow feature networks as well as the task specific head. The model is trained end-to-end. An overview over the models structure is given in figure 2. The model architecture, the reasoning for leveraging the *uncontextualized* embeddings for intelligibility, as well as the identifiability constraints are explained in detail below. In summary, we present a model that achieves identical performance to FT-Transformers (Gorishniy et al., 2021) while maintaining intelligibility with marginally more trainable parameters.

**Feature Encoding and Embedding**    Let $\mathcal{D} = \{(\boldsymbol{x}^{(i)}, y^{(i)})\}_{i=1}^{n}$ be the training dataset of size $n$ and let $y$ denote the target variable that can be arbitrarily distributed. Each input $\boldsymbol{x} = (x_1, x_2, \ldots, x_J)$ contains $J$ features. Let further $\boldsymbol{x} \equiv (\boldsymbol{x}_{cat}, \boldsymbol{x}_{num})$ denote the partition of the features into categorical

---

Please see Appendix A for an in depth literature review.

and numerical (continuous) features that constitute the whole feature vector $\boldsymbol{x}$. Further, let $x_{j(cat)}^{(i)}$ denote the $j$-th categorical feature of the $i$-th observation, and hence $x_{j(num)}^{(i)}$ denote the $j$-th numerical feature of the $i$-th observation.

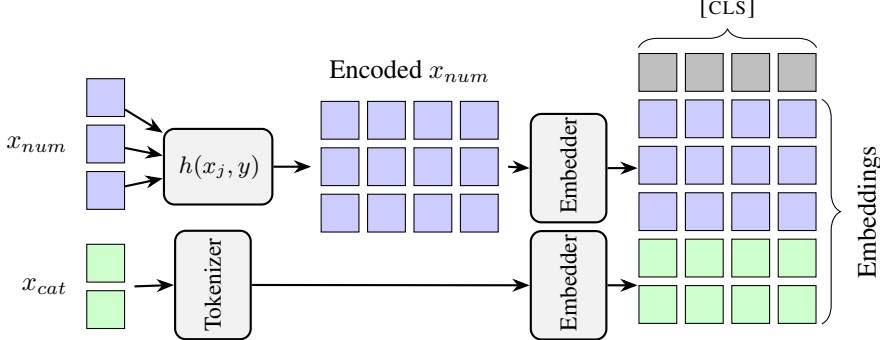

Figure 1: Feature Encoding. The numerical features are independently encoded $(h(x_j, y))$ and afterwards passed through an embedding layer. The categorical feature are tokenized and also passed through an embedding layer.

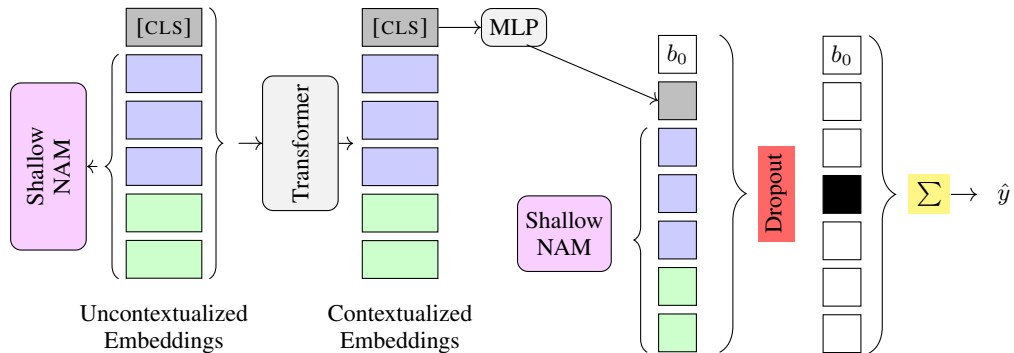

(a) Generation of the embeddings. A [CLS] token is appended to the uncotenxtualized embeddings before being passed through transformer blocks. The uncontextualized embeddings are also inputs to the single-layer shallow feature networks.

(b) The contextualized [CLS] token is passed through a task specific MLP head. The output of the shallow feature networks as well as the output of the MLP is passed through a dropout layer and summed to create the model output.

Figure 2: Training procedure of the proposed model structure. The architecture is conceptually very similar to FT-Transformer but allows to identify for marginal feature effects. Note, that feature dropout is only applied during training.

To leverage meaningful shallow networks, all elements of the numerical features in $x_{num}$ are encoded into a target-aware higher-dimensional space, using either target-aware one-hot encodings or piecewise linear encodings (PLE) (Gorishniy et al., 2022). Thus, for all numerical features $x_{j(num)} \in \mathbb{R}$, $x_{j(num)}$ is encoded such that it is either an element of $\mathbb{N}^{T_j}$ (one-hot) or in $\mathbb{R}^{T_j}$ (PLE), with a feature specific, target dependent encoding function $h_j(\boldsymbol{x}_{j(num)}, y)$. Decision trees are used for all $h_j(\cdot)$. We orientate on Gorishniy et al. (2022) and denote the encoded feature as $\boldsymbol{z}_{j(\text{num})}$ with entries

**One-hot encoding**

$$z_{j(\text{num})}^t = \begin{cases} 0 & \text{if } x < b_t, \\ 1 & \text{if } x \geq b_t. \end{cases}$$

**PLE**

$$z_{j(\text{num})}^t = \begin{cases} 0 & \text{if } x < b_{t-1}, \\ 1 & \text{if } x \geq b_t, \\ \frac{x - b_{t-1}}{b_t - b_{t-1}} & \text{else.} \end{cases}$$

where $b_t$ denote the decision boundaries from the decision trees. The dimension of the encoding $T_j$ depends on the feature, and not all features are necessarily mapped to the same dimension.

Following classical tabular transformer architectures, $\mathbf{E}_j(\cdot)$ represents the embedding function for feature $j$. Depending on the feature type, $\mathbf{E}_j$ embeds into the embedding space as follows: $\mathbf{E}_j : \mathbb{R}^{T_j} \to \mathbb{R}^e$ for numerical PLE encoded features, $\mathbf{E}_j : \mathbb{N}^{T_j} \to \mathbb{R}^e$ for numerical one-hot encoded features, and $\mathbf{E}_j : \mathbb{N} \to \mathbb{R}^e$ for categorical features. Categorical features are fed through standard embedding layers, and numerical features are passed through single linear layers as also done by Gorishniy et al. (2021).

It is worth noting that the embedding dimensionality can be chosen arbitrarily and can be smaller or larger than the dimensionality of the encoded features.

**From Uncontextual Embeddings to Marginal Predictions**  To understand the potential of *uncontextualized* embeddings as direct representations of raw input features in non-textual data settings, our investigation first examines their role within tabular data models. Importantly, using target aware encodings for preprocessing introduces non-linearity to numerical features, similar to neural spline expansions (Luber et al., 2023).

State-of-the-art language models, leveraging the Transformer architecture (Vaswani et al., 2017) first create context-insensitive input token representations. In Natural Language Processing, these are the raw, *uncontextualized* word embeddings. Subsequently, they compute $L$ layers of context-dependent representations, finally resulting in contextualized embeddings of the raw word representations (Peters et al., 2018). The proposed NAMformer architecture employs these *uncontextualized* embeddings directly to generate identifiable marginal feature predictions in a tabular context, necessitating a thorough analysis of their capability and effectiveness. By leveraging the *uncontextualized* embeddings, we aim to evaluate their utility in the proposed model architecture. This exploration is critical as it may reveal that raw, minimally processed embeddings can sufficiently capture and represent the essential characteristics of the features, potentially simplifying the model architecture while maintaining high predictive accuracy and interpretability.

In the context of tabular transformer networks, which do not employ positional encodings, the nature of *context* differs significantly from that in transformer models trained on textual data. The *context* that is added in the transformer layers, consists of the feature interactions (Huang et al., 2020). The *uncontextualized*, raw embeddings, however, are seldom used and are merely a byproduct of the model architecture. Since the input data for numerical features is not tokenized, *token identifiability* (Brunner et al., 2019) directly applies to the tabular input data. To confirm that the *uncontextualized* embeddings are not compromised by the subsequent layers during training, a straightforward experiment is conducted: First, we train a tabular transformer model using the California housing dataset[1]. Second, we extract the *uncontextualized* embeddings and analyze how well the true

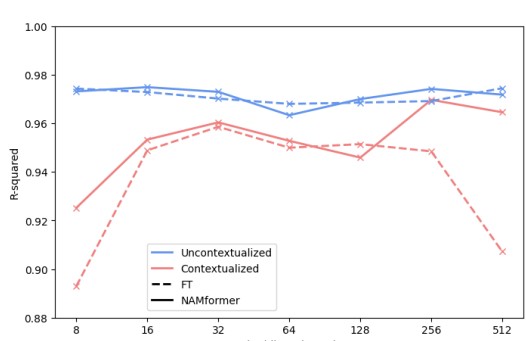

Figure 3: Average $R^2$ values over all 9 features. The decision trees are fit, using either the *uncontextualized* or the contextualized embeddings as training data and the true features as target variables.

feature data can be recognized. We follow Brunner et al. (2019) and train a set of $J$ simple decision trees $dt^j : \mathbb{R}^e \to \mathbb{R}$ on the embedding of every feature $x_j$. The true inputs, $x_j$ serve as the target variables, whereas the *uncontextualized* embeddings $\mathbf{E}_j(\boldsymbol{x}_j)$ are the training features. To put it differently, for each feature, we investigate how well it can be predicted with its contextualized or uncontextualized embedding. We report the $R^2$ values and find that *token identifiability* directly transfers to tabular input data and that *uncontextualized* embeddings are nearly perfect representations of the true data, with $R^2$ values of $\geq 0.96$ for different embedding sizes. For further details on the experimental setup, see the appendix.

---

[1]See Appendix D for details on the used datasets

**Additivity Constraint** Since we have verified that the *uncontextualized* embeddings almost perfectly preserve the original single feature information, we use them to include marginal feature predictions in the model. To achieve this, the *uncontextualized* embeddings are passed through single layer neural networks, similar to neural splines (e.g. (Luber et al., 2023; Rügamer et al., 2021). Subsequently, we make use of a simple additivity constraint following GAMs. Given a link function $g(\cdot)$ that connects the linear predictor to the expected value of the response variable, accommodating different types of data distributions, a GAM in its fundamental form can be expressed as follows:

$$g(\mathbb{E}\left[y\,|\,x_1, x_2, \ldots x_J\right]) = \beta_0 + \sum_{j=1}^{J} f_j(x_j), \tag{1}$$

where $\beta_0$ denotes the global intercept or bias term and $f_j : \mathbb{R} \to \mathbb{R}$ denote the univariate shape functions corresponding to input feature $x_j$ and capturing the feature main effects.

Let then $f_j^\epsilon : \mathbb{R}^e \to \mathbb{R}$ represent the shape function for the $j$-th feature's uncontextualized embedding. Let $H(\cdot)$ represent a sequence of transformer layers that take as input a sequence of all the uncontextualized embeddings $(\mathbf{E}_j(x_j))_{j=1}^{J}$ and output a sequence of contextualized embedding, such that $(\mathbf{\Xi}_j)_{j=1}^{J} = H((\mathbf{E}_j(x_j))_{j=1}^{J})$. For simplicity, we denote the uncontextualized embeddings as $\epsilon_j$ and the contextualized embeddings as $\mathbf{\Xi}_j$, where $\mathbf{\Xi}_j = H(\epsilon_1, \epsilon_2, \ldots \epsilon_J)_j$. Appending a [CLS] token to the uncontextualized embeddings additionally allows for interpreting attention weights and emulating feature importance (Gorishniy et al., 2021). Let $G$ further represent the MLP for processing the contextualized embeddings (or the [CLS] token embedding).

The final model combines the individual transformed uncontextualized embeddings $\epsilon_j$ for each feature, along with the contextual embeddings $\mathbf{\Xi}_j$. This setup ensures that both individual feature effects (via shape functions) and global contextual interactions via processing of the contextual embeddings are accounted for and interpretable in the model's output

$$g(\mathbb{E}\left[y|x_1, x_2, \ldots x_J\right]) = \beta_0 + \sum_{j=1}^{J} f_j^\epsilon(\epsilon_j) + G(\mathbf{\Xi}_j). \tag{2}$$

Using target-aware encodings for numerical features allows to use shallow, single-layer networks for the individual shape function $f_j^\epsilon : \mathbb{R}^e \to \mathbb{R}$ and thus account for interpretable marginal feature effects by only increasing the total number of parameters by $J \times e$.

## 2.1 IDENTIFIABILITY VIA FEATURE DROPOUT

For simplicity, we change notation and assume an additive model, such as the NAMformer, that has the following additive predictor:

$$\hat{\eta} = \beta_0 + \sum_{j=1}^{J} f_j(x_j) + f_{J+1}(x_1, x_2, \ldots, x_J), \tag{3}$$

where the marginal effects are modelled in separate networks, $f_j : \mathbb{R} \to \mathbb{R}$ and all interaction effects are jointly modeled in network, $f_{J+1} : \mathbb{R}^J \to \mathbb{R}$. This simplifies our proposed model architecture, but is transferable one to one.

Further, assume the model is fitted with shape function dropout and a risk of (at most) $R$. The loss function $\mathcal{L}$ is induced by the choice of the link function $g$ and distributional assumption in 2. Shape function dropout, introduced by (Agarwal et al., 2021), randomly drops out one or several features and their predictions in an additive model and is the main mechanism to ensure identifiability for NAMs. Here, let $\mathbf{w} \in \{0, 1\}^{J+1}$ denote the shape function dropout vector leading to the following risk:

$$\mathbb{E}_{\mathbf{x}, y \sim P^{\mathcal{D}}}\left[\mathbb{E}_{\mathbf{w} \sim P^{\mathbf{w}}}\left[\mathcal{L}\left(\beta_0 + \sum_{j=1}^{J} w_j f_j(x_j) + w_{J+1} f_{J+1}(x_1, x_2, \ldots, x_J), y\right)\right]\right] = R, \tag{4}$$

where $P^{\mathcal{D}}$ denotes the distribution of the data and $P^{\mathbf{w}}$ denotes the distribution over feature dropout weights.

Now, with Kronecker delta $\delta_{jk}$, let $\tilde{\mathbf{w}}_k = (\delta_{jk})_{j=1}^{J+1}$ be the dropout weight vector that drops out everything except for the effect of $f_k$, i.e. $\tilde{\mathbf{w}}_k$ has a one exactly at the $k$-th positions and zeros everywhere else. Then

$$R = \mathbb{E}_{\mathbf{x},y \sim P^{\mathcal{D}}} \left[ \mathcal{L} \left( \beta_0 + f_k(x_k), y \right) \right] p(\tilde{\mathbf{w}}_k) + R_{\tilde{\mathbf{w}}_{-k}}(1 - p(\tilde{\mathbf{w}}_k)),$$

where $R_{\tilde{\mathbf{w}}_{-k}} = R - \mathbb{E}_{\mathbf{x},y \sim P^{\mathcal{D}}} \left[ \mathcal{L} \left( \beta_0 + f_k(x_k), y \right) \right]$ is difference between the overall risk and the risk associated with $\tilde{\mathbf{w}}_{-k}$. Hence,

$$\mathbb{E}_{x_k,y \sim P^{\mathcal{D}}} \left[ \mathcal{L} \left( \beta_0 + f_k(x_k), y \right) \right] = \frac{R - R_{\tilde{\mathbf{w}}_{-k}}(1 - p(\tilde{\mathbf{w}}_k))}{p(\tilde{\mathbf{w}}_k)}. \tag{5}$$

Assuming a general distance-based loss $\mathcal{L}(y, \hat{y}) = g_{\mathcal{L}}(y - \hat{y})$ for a convex function $g_{\mathcal{L}}$, one obtains with Jensen's inequality:

$$\begin{aligned}
\mathbb{E}_{x_k,y \sim P^{\mathcal{D}}} \left[ \mathcal{L} \left( \beta_0 + f_k(x_k), y \right) \right] &= \mathbb{E}_{x_k,y \sim P^{\mathcal{D}}} \left[ g_{\mathcal{L}} \left( \beta_0 + f_k(x_k) - y \right) \right] \\
&= \mathbb{E}_{x_k} \left[ \mathbb{E}_{y|x_k} \left[ g_{\mathcal{L}} \left( \beta_0 + f_k(x_k) - y \right) | x_k \right] \right] \geq \mathbb{E}_{x_k} \left[ g_{\mathcal{L}} \left( \mathbb{E}_{y|x_k} \left[ \beta_0 + f_k(x_k) - y | x_k \right] \right) \right] \\
&= \mathbb{E}_{x_k} \left[ g_{\mathcal{L}} \left( \beta_0 + f_k(x_k) - \mathbb{E}_{y|x_k} \left[ y | x_k \right] \right) \right] = \mathbb{E}_{x_k} \left[ \mathcal{L} \left( \beta_0 + f_k(x_k), \mathbb{E}_{y|x_k} \left[ y | x_k \right] \right) \right]. \tag{6}
\end{aligned}$$

Most common regression loss-functions such as the $L^p$ losses, the Huber loss or the Pinball loss are all distance based loss function of the form assumed above. Furthermore, an analogous argument can be made in the binary classification case with a margin-based binary (classification) loss of the form $\mathcal{L}(y, \hat{s}) = h_{\mathcal{L}}(y \cdot \hat{s})$ with labels $y \in \{-1, 1\}$ and $\hat{s} \in \mathbb{R}$ the output of a scoring classifier (see Appendix C).

In summary, it is shown for broad classes of regression and classification losses $\mathcal{L}$ that we can identify the true marginal effect $\mathbb{E}_{y|x_k} \left[ y | x_k \right]$ with the following error, measured in terms of the original loss function $\mathcal{L}$:

$$\mathbb{E}_{x_k} \left[ \mathcal{L} \left( \beta_0 + f_k(x_k), \mathbb{E}_{y|x_k} [y | x_k] \right) \right] \leq \frac{R - R_{\tilde{\mathbf{w}}_{-k}}(1 - p(\tilde{\mathbf{w}}_k))}{p(\tilde{\mathbf{w}}_k)} \tag{7}$$

Our upper bound thus shows that minimizing the ratio between, first, the difference between overall risk and the risk associated with $\tilde{\mathbf{w}}_{-k}$ and, second, the dropout probability for only keeping the $k$-th vector implies a low risk in identifying marginal effects.

When the risk $R$ is uniformly distributed among all values of $\tilde{\mathbf{w}}_k$, such that $R_{\tilde{\mathbf{w}}_{-k}} = R \cdot (1 - p(\tilde{\mathbf{w}}_k))$, one gets the following bound on the expected error with respect to the ground-truth effect:

$$\mathbb{E}_{x_k} \left[ \mathcal{L} \left( \beta_0 + f_k(x_k), \mathbb{E}_{y|x_k} [y | x_k] \right) \right] \leq \frac{R \cdot (1 - (1 - p(\tilde{\mathbf{w}}_k))^2)}{p(\tilde{\mathbf{w}}_k)} = R \cdot (2 - p(\tilde{\mathbf{w}}_k)) \leq 2R \tag{8}$$

Please note, that the case where the risk is perfectly uniformly distributed among all values of $\tilde{\mathbf{w}}_k$ is unlikely in practice and represents an upper bound.

## 3 ABLATION

**Simulation Study** First, it is analyzed how well the proposed model can identify marginal feature effects, also when complex feature interactions are present. NAMFormer is compared with other, neural (Agarwal et al., 2021) and decision tree based intelligible models (Nori et al., 2019), as well as a simple linear regression model and a GAM (Hastie, 2017). Multiple datasets are simulated with a

normally distributed target variable. Each dataset follows a straightforward data generating process where $y = \sum_{j=1}^{J} s_j(x_j) + \prod x_j + \varepsilon$, such that $s_j$ are the marginal feature effects. See appendix E for further information on the data generating process. Subsequently, each model is fit on the dataset and we analyze the marginal feature predictions. Explainable Boosting Machines (EBM) (Nori et al., 2019) are fit using the default hyperparameters. GAMs are fit using cubic splines with 25 knots each. NAMs follow the architecture established from Radenovic et al. (2022) and Dubey et al. (2022). For NAMformer, we use embedding sizes of 32, 4 layers, 2 heads, attention dropout of 0.3 and feed forward dropout of 0.3. For NAMs and NAMformer we use identical feature dropout probability of 0.1, since the shown identifiability is also a core feature of NAMs. Additionally, we compare, target aware one-hot encodings with 150 bins, PLE encodings with 25 bins and standardization of numerical features for NAMformer[2].

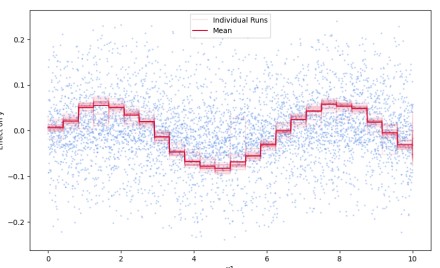 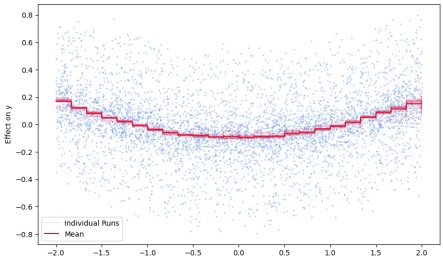

Figure 4: Marginal feature predictions for a simple simulated example with 4 variables and the described data generating process. Over 25 runs, and with only 25 bins, NAMformer accurately identifies the marginal effects.

Subsequently, we analyze the marginal feature predictions and calculate the $R^2$ values for each marginal feature prediction with respect to the true data generating function. Table 1 shows the averaged results over all effects.

Table 1: Average $R^2$ values over marginal feature effects for different datasets. With increasing index, the number of effects as well as the complexity of the data increase. Larger values are better. The gray $\pm$ values denote the standard deviation among the calculated $R^2$ value for the different marginal effects. *oh* denotes one-hot encoded features, *st* standardized features and *ple* piecewise linear encodings.

| | Number of marginal effects | | | | | | |
|---|---|---|---|---|---|---|---|
| Model | 3 | 4 | 5 | 6 | 7 | 8 | 9 |
| Linear | 0.467 ±0.41 | 0.251 ±0.67 | 0.220 ±0.62 | 0.238 ±0.59 | 0.124 ±0.63 | 0.034 ±0.68 | 0.092 ±0.68 |
| GAM | 0.800 ±0.37 | 0.534 ±0.76 | 0.466 ±0.73 | 0.500 ±0.69 | 0.356 ±0.76 | 0.257 ±0.81 | 0.299 ±0.79 |
| EBM | 0.797 ±0.37 | 0.531 ±0.76 | 0.464 ±0.73 | 0.500 ±0.69 | 0.371 ±0.73 | 0.266 ±0.79 | 0.331 ±0.74 |
| EB$^2$M | 0.797 ±0.37 | 0.531 ±0.76 | 0.464 ±0.73 | 0.500 ±0.69 | 0.371 ±0.73 | 0.266 ±0.79 | 0.331 ±0.74 |
| NAM | 0.741 ±0.35 | 0.653 ±0.39 | 0.611 ±0.39 | 0.648 ±0.39 | 0.629 ±0.40 | 0.477 ±0.61 | 0.507 ±0.59 |
| Hi-NAM | 0.801 ±0.36 | 0.556 ±0.75 | 0.577 ±0.63 | 0.676 ±0.47 | 0.597 ±0.50 | 0.526 ±0.64 | 0.658 ±0.57 |
| NAMformer$_{st}$ | **0.865** ±0.23 | 0.662 ±0.57 | 0.596 ±0.53 | 0.781 ±0.28 | 0.605 ±0.43 | 0.631 ±0.58 | 0.770 ±0.56 |
| NAMformer$_{oh}$ | 0.826 ±0.11 | 0.877 ±0.13 | 0.737 ±0.14 | 0.837 ±0.09 | **0.922** ±0.13 | **0.722** ±0.25 | **0.826** ±0.23 |
| NAMformer$_{ple}$ | 0.806 ±0.40 | **0.918** ±0.15 | **0.879** ±0.17 | **0.867** ±0.11 | 0.909 ±0.10 | 0.617 ±0.59 | 0.756 ±0.56 |

We find that NAMformer can accurately identify marginal effects, even in the presence of higher-order feature interactions. Interestingly, using target aware encodings is also beneficial for feature identifiability in the NAMformer. Additionally, while NAMs implementing the same identifiability regularizer can also identify the marginal effects, their performance diminishes as more interactions are introduced. Furthermore, NAMs exhibit significantly larger standard deviations among the $R^2$ values for individual effects compared to NAMformer, which identifies all effects with smaller deviation.

**Comparison to FT-Transformer** Since the proposed architecture closely follows the FT-Transformer architecture from Gorishniy et al. (2021), we first compare whether introducing identifi-

---

[2]See appendix E for the experimental details.

Table 2: Comparison between NAMformer and FT-Transformer with identical hyperparameters on different datasets. 5-fold cross validation was performed. The average performances for both models are not out of the bounds of the standard deviations over the 5-folds. Hence, we find that NAMformer, while producing identifiable marginal effects performs as good as FT-Transformer.

| Model | CH $\downarrow$ | MU $\downarrow$ | DM $\downarrow$ | HS $\downarrow$ | AD $\uparrow$ | BA $\uparrow$ | SH $\uparrow$ | FI $\uparrow$ |
|---|---|---|---|---|---|---|---|---|
| NAMformer$_{st}$ | 0.235 | 0.798 | 0.021 | 0.131 | 0.908 | 0.953 | 0.862 | 0.788 |
| | $\pm0.013$ | $\pm0.351$ | $\pm0.001$ | $\pm0.021$ | $\pm0.003$ | $\pm0.006$ | $\pm0.007$ | $\pm0.021$ |
| FT-T$_{st}$ | 0.227 | 0.780 | 0.023 | 0.127 | 0.908 | 0.960 | 0.861 | 0.790 |
| | $\pm0.011$ | $\pm0.323$ | $\pm0.002$ | $\pm0.018$ | $\pm0.002$ | $\pm0.010$ | $\pm0.009$ | $\pm0.010$ |
| NAMformer$_{oh}$ | 0.220 | 0.801 | 0.022 | 0.162 | 0.903 | 0.901 | 0.825 | 0.766 |
| | $\pm0.007$ | $\pm0.379$ | $\pm0.001$ | $\pm0.021$ | $\pm0.006$ | $\pm0.022$ | $\pm0.006$ | $\pm0.013$ |
| FT-T$_{oh}$ | 0.225 | 0.901 | 0.024 | 0.158 | 0.899 | 0.644 | 0.820 | 0.763 |
| | $\pm0.007$ | $\pm0.417$ | $\pm0.002$ | $\pm0.029$ | $\pm0.007$ | $\pm0.144$ | $\pm0.010$ | $\pm0.010$ |
| NAMformer$_{ple}$ | 0.206 | 0.642 | 0.020 | 0.127 | 0.912 | 0.945 | 0.858 | 0.789 |
| | $\pm0.007$ | $\pm0.241$ | $\pm0.001$ | $\pm0.016$ | $\pm0.002$ | $\pm0.007$ | $\pm0.005$ | $\pm0.013$ |
| FT-T$_{ple}$ | 0.197 | 0.834 | 0.023 | 0.129 | 0.910 | 0.944 | 0.858 | 0.789 |
| | $\pm0.011$ | $\pm0.420$ | $\pm0.001$ | $\pm0.022$ | $\pm0.005$ | $\pm0.020$ | $\pm0.007$ | $\pm0.009$ |

able marginal feature networks hampers predictive power compared to classical FT-Transformers. We fit both models with identical transformer architectures and use the same feature encoding and preprocessing methods for both models. We perform 5-fold cross-validation and compare mean squared error values on 4 regression datasets and Area under the curve (AUC) on 4 binary classification datasets. See appendix F for details on the experimental setup. Notably, we do not find that either model performs better or worse, as no model achieves significantly different performances with respect to the cross validation results. This strengthens our hypothesis, that adding marginally identifiable networks with minimally more parameters ($J \times e < 5000$ for all datasets) does not harm predictive performance at all.

## 4 EXPERIMENTS

NAMformer is compared with several interpretable as well as black-box models using 11 regression and 4 classification datasets. All data splits and the descriptions can be found in the Appendix D. Hyperparameters are tuned for all models, orientated on the benchmarks performed by Gorishniy et al. (2021). See Appendix G for details. Additionally to GAMs, EBMs and NAMs, we fit a Hi-NAM (Kim et al., 2022), a NAM that incorporates a single MLP fit on all features and thus captures all (higher-order) feature interactions. Additional results are reported in Appendix B.

**Results** The results for interpretable models are reported in Table 3. NAMformer performs (shared) best for 9 out of 15 datasets. Additionally, the results demonstrate strong support for EBMs, with both, EBMs and EB$^2$Ms performing strongly. While NAMs perform only marginally better than classical GAMs and on some datasets even worse, Hi-NAMs also perform strongly, especially on regression tasks. Note, that all models are fine-tuned and hence we achieve different results than for the identically implemented architectures from table 2.

Computing average ranks among all interpretable models among all tasks also reveals that NAMformer is the best performing model on average, followed by EB$^2$M. See Table 5 in Appendix B.

For black-box models, NAMformer are compared to classical MLPs, XGBoost and FT-Transformer. We use standardized preprocessed features for the FT-Transformer since we found them to perform best in our initial experiment (Table 2) [3]. Overall, the experiments confirm the results from Gorishniy et al. (2021) that FT-Transformer can outperform XGBoost on certain datasets.

---

[3]Note, that we tune the hyperparamters of NAMformer and FT-Transformer separately and thus get different results than those from table 2.

Table 3: Results for interpretable models. For regression problems (CH, MU, DM, HS, AV, GS, K8, P32, MH, BH, SG), MSE values are reported. For binary classification problems (AD, BA, SH, FI), the area under the curve (AUC) as well as the accuracy (in gray) are reported.

**Regression Results (MSE ↓)**

| Model | CH ↓ | MU ↓ | DM ↓ | HS ↓ | AV ↓ | GS ↓ | K8 ↓ | P32 ↓ | MH ↓ | BH ↓ | SG ↓ |
|---|---|---|---|---|---|---|---|---|---|---|---|
| Linear | 0.370 | 0.726 | 0.115 | 0.333 | 0.700 | 0.366 | 0.580 | 0.843 | 0.295 | 0.025 | 0.445 |
| GAM | 0.288 | 0.747 | 0.066 | 0.267 | 0.287 | 0.228 | 0.557 | 0.909 | 0.157 | 0.023 | 0.273 |
| EBM | 0.195 | 0.703 | 0.023 | 0.205 | 0.050 | 0.079 | 0.411 | 0.395 | 0.096 | 0.033 | 0.272 |
| EB$^2$M | 0.194 | 0.695 | 0.023 | 0.201 | 0.049 | 0.079 | 0.409 | 0.388 | **0.099** | 0.026 | **0.263** |
| NAM | 0.306 | 0.735 | 0.069 | 0.451 | 0.372 | 0.235 | 0.927 | 1.049 | 0.181 | 0.025 | 0.399 |
| Hi-NAM | 0.194 | 0.718 | **0.022** | **0.132** | 0.135 | **0.034** | **0.076** | 0.435 | 0.102 | 0.128 | 0.278 |
| NAMformer | **0.173** | **0.668** | **0.022** | 0.148 | **0.023** | 0.051 | 0.108 | 0.397 | **0.095** | **0.022** | 0.270 |

**Classification Results (AUC ↑ and Accuracy in gray)**

| | AD ↑ | BA ↑ | SH ↑ | FI ↑ |
|---|---|---|---|---|
| Linear | 0.852 82.4% | 0.871 88.6% | 0.764 81.5% | 0.754 69.3% |
| GAM | 0.913 85.9% | 0.911 90.1% | 0.855 86.4% | 0.779 70.9% |
| EBM | **0.927** 87.3% | **0.931** 90.8% | 0.868 86.8% | **0.783** 70.8% |
| EB$^2$M | **0.927** 87.3% | **0.931** 90.8% | 0.870 86.3% | **0.783** 70.8% |
| NAM | 0.910 85.3% | 0.901 89.4% | 0.853 86.2% | 0.776 70.0% |
| Hi-NAM | 0.910 85.4% | 0.911 89.7% | 0.858 86.5% | 0.779 70.3% |
| NAMformer | **0.927** 87.2% | **0.931** 90.8% | **0.871** 86.5% | 0.780 70.7% |

Table 4: Results for black-box models. For regression problems (CH, MU, DM, HS) mse values are reported, for binary classification problems (AD, BA, SH, FI) the area under the curve as well as the accuracy (in gray) are reported.

| Model | CH ↓ | MU ↓ | DM ↓ | HS ↓ | AD ↑ | BA ↑ | SH ↑ | FI ↑ |
|---|---|---|---|---|---|---|---|---|
| XGB | **0.159** | 0.728 | 0.018 | 0.163 | **0.927** 87.3% | **0.933** 90.5% | 0.868 86.3% | **0.781** 70.1% |
| FT-T | 0.184 | 0.688 | **0.017** | **0.111** | 0.915 85.9% | 0.929 90.2% | 0.870 86.3% | 0.777 70.1% |
| MLP | 0.195 | 0.725 | 0.018 | 0.164 | 0.908 89.8% | 0.913 85.5% | 0.862 86.5% | 0.771 70.0% |
| NAMformer | 0.173 | **0.668** | 0.022 | 0.148 | **0.927** 87.2% | 0.931 90.8% | **0.871** 86.5% | 0.780 70.7% |

# 5 CONCLUSION

We present NAMformer, an effective adaptation to the FT-Transformer architecture. We can effectively incorporate marginal feature effects and show theoretical justification of our approach. With minimally more parameters compared to the FT-Transformer architecture, NAMformer achieve identical performance while also generating identifiable marginal feature predictions. The reasoning for including an additivity constraint and fitting shallow feature networks in tabular transformers is thus that without loss of generalizability and without loss of performance, we can get an interpretable model at the cost of - depending on the embedding size and the number of features - marginally more parameters. Our theoretical justification of identifiable marginal feature effects is also seamlessly applicable to models incorporating unstructured data (Rügamer et al., 2023; Reuter et al., 2024). Therefore, the achieved results are a further step into intelligible deep learning models beyond tabular data analysis.

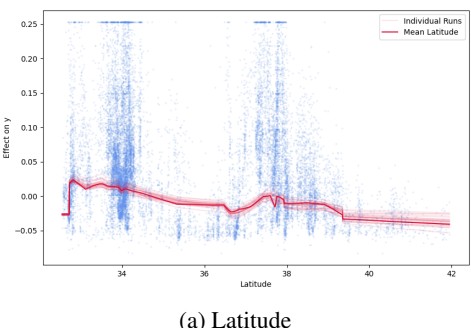

(a) Latitude

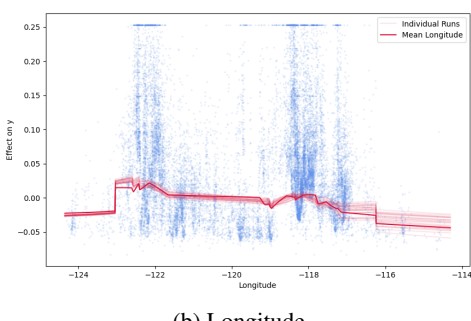

(b) Longitude

Figure 5: Marginal feature predictions from 25 trained NAMformer models on the California housing datasets for the variables "latitude" and "longitude" using PLE encodings.

# 6 LIMITATIONS

The interpretability of NAMformer, while significantly better than that of black-box models, still does not match the inherent statistical inference capabilities of classical GAMs. True interpretability in the form of significance statistics is still a problem for further research.

Additionally, this paper solely focuses on single marginal feature effects. While we account for high-order feature interactions, we do not explicitly account for, e.g., second order feature interactions as models like $EB^2M$ do. Hence, interesting interaction effects as the ones between, e.g., longitude and latitude are not specifically accounted for. However, the introduced identifiability constraint seamlessly enables to account for any amount and order of feature interactions that one wants to account for. Simple feature interaction networks can be easily incorporated and fit on a combination of the *uncontextualized* embeddings.

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

## A  LITERATURE REVIEW

This manuscript can be categorized into two main areas of literature: *tabular deep learning* and *additive interpretable modeling*, the latter being largely inspired by classical statistical models. An additive model, as the name suggests, learns marginal feature effects and derives its final prediction by summing over these effects (e.g., see Hastie (2017)). These models originate from simple linear regression, but instead of relying on linear effects, generalized additive models (GAMs) allow for the learning of more complex relationships through shape functions, as shown in Equation 1.

A key aspect of additive models is that they allow interpretation of marginal feature effects, which quantify the isolated impact of each individual feature on the prediction while holding all other features constant. Marginal effects provide a clear, interpretable mapping of feature-to-outcome relationships, making them especially valuable in domains such as healthcare, finance, and policy-making, where understanding why a model makes a specific prediction is critical (e.g. (Hastie & Tibshirani, 1995; Barrio et al., 2013; Mize et al., 2019)). By offering transparency into the model's reasoning, interpretable marginal effects allow practitioners to validate predictions, build trust in the model, and gain actionable insights for decision-making.

Classical GAMs (e.g., (Hastie, 2017; Wood, 2017)) often use splines for basis expansions. For an introduction, see (Fahrmeir et al., 2013). This approach offers significant advantages, particularly in terms of interpretability and intelligibility. However, detecting complex feature effects, especially those involving interactions, can be challenging for spline-based models. This limitation has motivated the development of Neural Additive Models (NAMs) (Agarwal et al., 2021), which replace splines with shape functions modeled via multi-layer perceptrons (MLPs) optimized through gradient descent.

Extensions of NAMs, such as those proposed in (Thielmann et al., 2023; 2024; Luber et al., 2023; Kim et al., 2022; Chang et al., 2021), build on this simple additive modeling concept. Although these models outperform classical GAMs, they often lag behind the performance of models like XGBoost or FT-Transformers (Gorishniy et al., 2021).

Tabular deep learning models, such as the FT-Transformer, do not impose the additivity constraint from Equation 1. As a result, they can effectively capture higher-order feature interactions. The underlying approach is straightforward: all features are passed jointly through the architecture. In the case of FT-Transformers, features are first embedded into a higher-dimensional space via embedding layers (Gorishniy et al., 2022), then processed through the transformer blocks. Finally, the outputs are pooled and passed through a task-specific model head to derive the final prediction.

Although tabular deep learning models and (tabular) additive models are closely related, to the best of our knowledge, no existing model achieves both *interpretable marginal effect modeling* and the performance of state-of-the-art models like XGBoost or FT-Transformers.

## B  ADDITIONAL RESULTS

Average ranks among all models and all 15 datasets. HPO is performed as reported in the main text.

Table 5: Average ranks and rank standard deviations of models.

| Model | Average Rank | Rank Std Dev |
|---|---|---|
| NAMformer | **1.700** | 0.678 |
| EB$^2$M | 2.467 | 1.024 |
| EBM | 3.100 | 1.114 |
| Hi-NAM | 3.567 | 1.740 |
| GAM | 4.800 | 1.030 |
| NAM | 6.000 | 0.796 |
| Linear | 6.367 | 1.040 |

The initial results on 8 datasets, 4 regression and 4 classification, largely inspired from studies such as Agarwal et al. (2021); Thielmann et al. (2023); Chang et al. (2021) are shown below in table 6

Table 6: Results for interpretable models. For regression problems (CH, MU, DM, HS) mse values are reported, for binary classification problems (AD, BA, SH, FI) the area under the curve as well as the accuracy in gray are reported.

| Model | CH ↓ | MU ↓ | DM ↓ | HS ↓ | AD ↑ | BA ↑ | SH ↑ | FI ↑ |
|---|---|---|---|---|---|---|---|---|
| Linear | 0.370 | 0.726 | 0.115 | 0.333 | 0.852 82.4% | 0.871 88.6% | 0.764 81.5% | 0.754 69.3% |
| GAM | 0.288 | 0.747 | 0.066 | 0.267 | 0.913 85.9% | 0.911 90.1% | 0.855 86.4% | 0.779 70.9% |
| EBM | 0.195 | 0.703 | 0.023 | 0.205 | **0.927** 87.3% | **0.931** 90.8% | 0.868 86.8% | **0.783** 70.8% |
| EB$^2$M | 0.194 | 0.695 | 0.023 | 0.201 | **0.927** 87.3% | **0.931** 90.8% | 0.870 86.3% | **0.783** 70.8% |
| NAM | 0.306 | 0.735 | 0.069 | 0.451 | 0.910 85.3% | 0.901 89.4% | 0.853 86.2% | 0.776 70.0% |
| Hi-NAM | 0.194 | 0.718 | **0.022** | **0.132** | 0.910 85.4% | 0.911 89.7% | 0.858 86.5% | 0.779 70.3% |
| NAMformer | **0.173** | **0.668** | **0.022** | 0.148 | **0.927** 87.2% | **0.931** 90.8% | **0.871** 86.5% | 0.780 70.7% |

Additional benchmarks on 7 additional regression datasets, taken from Fischer et al. (2023) are shown in table 7.

Table 7: Performance comparison of models across various metrics.

| Model | AV ↓ | GS ↓ | K8 ↓ | P32 ↓ | MH ↓ | BH ↓ | SG ↓ |
|---|---|---|---|---|---|---|---|
| Linear | 0.700 | 0.366 | 0.580 | 0.843 | 0.295 | 0.025 | 0.445 |
| GAM | 0.287 | 0.228 | 0.557 | 0.909 | 0.157 | 0.023 | 0.273 |
| EBM | 0.050 | 0.079 | 0.411 | 0.395 | 0.096 | 0.033 | 0.272 |
| EB²M | 0.049 | 0.079 | 0.409 | 0.388 | **0.099** | 0.026 | **0.263** |
| NAM | 0.372 | 0.235 | 0.927 | 1.049 | 0.181 | 0.025 | 0.399 |
| Hi-NAM | 0.135 | **0.034** | **0.076** | 0.435 | 0.102 | 0.128 | 0.278 |
| NAMformer | **0.023** | 0.051 | 0.108 | 0.397 | **0.095** | **0.022** | 0.270 |

## C  SHAPE FUNCTION DROPOUT FOR THE MSE-LOSS AND CLASSIFICATION LOSSES

In this section, we show how the result in section 2.1 can be made more precise in the case of an MSE loss and how it can be adapted to the case of margin-based classification losses. In general, we want to show

$$\mathbb{E}_{x_k} \left[ \mathcal{L} \left( \beta_0 + f_k(x_k), \mathbb{E}_{y|x_k}[y|x_k] \right) \right] \leq \mathbb{E}_{x_k, y \sim P^{\mathcal{D}}} \left[ \mathcal{L} \left( \beta_0 + f_k(x_k), y \right) \right]. \tag{9}$$

First, one obtains when assuming an MSE-Loss $\mathcal{L}(y, \hat{y}) = (y - \hat{y})^2$:

$$\mathbb{E}_{x_k, y \sim P^{\mathcal{D}}} \left[ \mathcal{L} \left( \beta_0 + f_k(x_k), y \right) \right] = \mathbb{E}_{x_k, y \sim P^{\mathcal{D}}} \left[ \left( \beta_0 + f_k(x_k) - y \right)^2 \right] =$$

$$\mathbb{E}_{x_k} \left[ \mathbb{E}_{y|x_k} \left[ \left( \beta_0 + f_k(x_k) - y \right)^2 | x_k \right] - \left( \mathbb{E}_{y|x_k} \left[ \beta_0 + f_k(x_k) - y | x_k \right] \right)^2 + \left( \mathbb{E}_{y|x_k} \left[ \beta_0 + f_k(x_k) - y | x_k \right] \right)^2 \right) =$$

$$\mathbb{E}_{x_k} \left[ \mathbb{V} \left[ y | x_k \right] \right] + \mathbb{E}_{x_k} \left[ \left( \beta_0 + f_k(x_k) - \mathbb{E}_{y|x_k}[y|x_k] \right)^2 \right] \tag{10}$$

Here, $\mathbb{E}_{x_k} \left[ \mathbb{V} \left[ y | x_k \right] \right] =: R_{x_k}$ is the irreducible error in predicting $y$ based on $x_k$. Thus one can obtain the following explicit expression for the risk:

$$\mathbb{E}_{x_k} \left[ \left( \beta_0 + f_k(x_k) - \mathbb{E}_{y|x_k}[y|x_k] \right)^2 \right] = \frac{R - R_{\tilde{\mathbf{w}}_{-k}}(1 - p(\tilde{\mathbf{w}}_k))}{p(\tilde{\mathbf{w}}_k)} - R_{x_k}$$

For margin-based binary classification losses of the form $\mathcal{L}(y, \hat{s}) = h_{\mathcal{L}}(y \cdot \hat{s})$, such that $y \in \{-1, 1\}$ and $\hat{s} \in \mathbb{R}$ is the output of a scoring classifier and $h_{\mathcal{L}}$ is convex, one obtains:

$$\mathbb{E}_{x_k, y \sim P^{\mathcal{D}}} \left[ \mathcal{L} \left( \beta_0 + f_k(x_k), y \right) \right] = \mathbb{E}_{x_k, y \sim P^{\mathcal{D}}} \left[ h_{\mathcal{L}} \left( y \cdot (\beta_0 + f_k(x_k)) \right) \right] =$$

$$\mathbb{E}_{x_k} \left[ \mathbb{E}_{y|x_k} \left[ h_{\mathcal{L}} \left( y \cdot (\beta_0 + f_k(x_k)) | x_k \right] \right] \geq \mathbb{E}_{x_k} \left[ h_{\mathcal{L}} \left( \mathbb{E}_{y|x_k} \left[ y \cdot (\beta_0 + f_k(x_k)) | x_k \right] \right) \right]$$

$$= \mathbb{E}_{x_k} \left[ h_{\mathcal{L}} \left( (\beta_0 + f_k(x_k)) \cdot \mathbb{E}_{y|x_k} \left[ y | x_k \right] \right) \right] = \mathbb{E}_{x_k} \left[ \mathcal{L} \left( \beta_0 + f_k(x_k), \mathbb{E}_{y|x_k} \left[ y | x_k \right] \right) \right]. \quad (11)$$

Note that here $\mathbb{E}_{y|x_k} [y|x_k] = 2\mathbb{P}(y = 1|x_k) - 1$. Thus for $\tilde{y} = \frac{y+1}{2} \in \{0, 1\}$, which is the 0-1 variant of the label $y$, and therefore $\tilde{\mathcal{L}}(\tilde{y}, \hat{s}) = h_{\mathcal{L}}((2\tilde{y} - 1) \cdot \hat{s})$, one gets

$$\mathbb{E}_{x_k, y \sim P^{\mathcal{D}}} \left[ \tilde{\mathcal{L}} \left( \beta_0 + f_k(x_k), \tilde{y} \right) \right] \geq \mathbb{E}_{x_k} \left[ \tilde{\mathcal{L}} \left( \beta_0 + f_k(x_k), \mathbb{E}_{y|x_k} \left[ \tilde{y} | x_k \right] \right) \right], \quad (12)$$

where $\mathbb{E}_{y|x_k} [\tilde{y}|x_k] = \mathbb{P}(\tilde{y} = 1|x_k)$, showing that the difference of the marginal effect of $x_k$ to the Bayes-Optimal classification model is bounded in this case.

Many common classification loss functions, such as the 0-1-loss, the Log-Loss, the Hinge Loss, the Exponential Loss can be expressed as a margin-based loss with a convex function $h_{\mathcal{L}}$.

# D  DATA

## D.1  DATASETS

Table 8: Details on datasets used in the experiments. The tasks are abbreviated as reg. for regression and cls. for (binary) classification.

| Abr | Name | # Total | # Train | # Val | # Test | # Num | # Cat | Task |
|-----|------|---------|---------|-------|--------|-------|-------|------|
| CH | California Housing | 20433 | 13076 | 3270 | 4087 | 8 | 1 | reg. |
| MU | Airbnb Munich | 6627 | 4240 | 1061 | 1326 | 5 | 4 | reg. |
| AB | Abalone | 4177 | 2672 | 669 | 836 | 7 | 1 | reg. |
| CU | CPU small | 8192 | 5242 | 1311 | 1639 | 12 | 0 | reg. |
| DM | Diamonds | 53940 | 34521 | 8631 | 10788 | 6 | 3 | reg. |
| HS | House Sales | 21613 | 13832 | 3458 | 4323 | 10 | 8 | reg. |
| AD | Adult | 48842 | 31258 | 7815 | 9769 | 5 | 8 | cls. |
| BA | Banking | 45211 | 28934 | 7234 | 9043 | 3 | 12 | cls. |
| SH | Churn Modeling | 10000 | 6400 | 1600 | 2000 | 8 | 2 | cls. |
| FI | FICO | 10459 | 6693 | 1674 | 2092 | 16 | 7 | cls. |
| AV | Auction Verification | 2043 1225 | 409 | 409 | 5 | 2 | reg. | |
| GS | Grid Stability | 10000 | 6000 | 2000 | 2000 | 12 | 0 | reg. |
| K8 | Kin8nm | 8192 | 4915 | 1638 | 1639 | 8 | 0 | reg. |
| P32 | Pumadyn32nh | 8192 | 4915 | 1638 | 1639 | 32 | 0 | reg. |
| MH | Miami Housing | 13932 | 8359 | 2786 | 1787 | 15 | 0 | reg. |
| BH | Brazilian Houses | 10692 | 6415 | 2138 | 2139 | 5 | 4 | reg. |
| SG | Space Ga | 3107 | 1864 | 621 | 622 | 6 | 0 | reg. |

### D.1.1  REGRESSION DATASETS

**California Housing**  The California Housing (CA Housing) dataset is a popular publicly available dataset. We obtained it from the UCI machine learning repository (Dua & Graff, 2017). We achieve similar results concerning the MSE for the models which were used e.g. in Agarwal et al. (2021), Thielmann et al. (2023) and Gorishniy et al. (2021). The dataset contains the house prices for California homes from the U.S. census in 1990. The dataset is comprised of 20433 and besides the target variable contains nine predictors. As described above, we additionally standard normalize the target variable. All other variables are preprocessed as described above.

**Munich**  For the AirBnB data, we orientate on Rügamer et al. (2023) and Thielmann et al. (2023) and used the data for the city of Munich. The dataset is publicly available and was taken from Inside AirBnB (http://insideairbnb.com/get-the-data/) on January 15, 2023. After cleaning, the dataset consist of 6627 observations. The target variable is the rental price.

**Diamonds**   The diamonds dataset is also taken from the UCI machine learning repository (Dua & Graff, 2017). We standard normalized the target variable and dropped out all rows that contained unknown values. A detailed description of all its features can be found here `https://www.openml.org/search?type=data&sort=runs&id=42225&status=active`

**House sales**   The dataset and its description can be found here `https://www.openml.org/search?type=data&status=active&id=42092`. We drop all rows that contain unknown values.

### D.1.2   CLASSIFICATION DATASETS

**FICO**   A detailed description of the features and their meaning is available at the Explainable Machine Learning Challenge (`https://community.fico.com/s/explainable-machine-learning-challenge`). The dataset is comprised of 10459 observations. We did not implement any preprocessing steps for the target variable.

**Churn**   This dataset contains information on the customers of a bank and the target variable is a binary variable reflecting whether the customer has left the bank (closed their account) or remains a customer. The data set can be found at Kaggle (`https://www.kaggle.com/datasets/shrutimechlearn/churn-modelling`) and is introduced by Kaggle (2019). After the processing described above, the set consists of 10000 observations, each with 10 features.

**Adult**   The adult dataset is another common benchmark dataset used in studies such as e.g. Grinsztajn et al. (2022); Arik & Pfister (2021); Ahamed & Cheng (2024). It is taken from `https://archive.ics.uci.edu/dataset/2/adult` and a detailed description can be found there.

**Banking**   A detailed description on the banking dataset can be found here `https://www.openml.org/search?type=data&status=active&id=44234`. It is also taken from the UCI machine learning repository (Dua & Graff, 2017).

## E   ABLATION STUDY

In the ablation study, we simulate a dataset consisting of 25,000 data points. We utilize a train-test split of 70% - 30% to evaluate the impact of various shape functions and categorical feature effects on the model's performance.

**Continuous Features**   We examine the following set of shape functions to model the continuous features. All x variables are uniformly distributed between 0 and 1 and independently sampled. The functions are designed to introduce a variety of nonlinear transformations:

- Linear function: $s_1(x) = 3x$
- Quadratic function: $s_2(x) = (x-1)^2$
- Sinusoidal function: $s_3(x) = \sin(5x)$
- Exponential root function: $f_4(x) = \sqrt{\exp(x)}$
- Absolute deviation: $s_5(x) = |x-1|$
- Sinusoidal deviation: $s_6(x) = |x - \sin(5x)|$
- Signed root function: $s_7(x) = \text{sign}(x) \cdot \sqrt{|x|}$
- Exponential-polynomial function: $s_8(x) = 2^x - x^2$
- Cubic polynomial: $s_9(x) = x^3 - 3x$
- Exponential increment: $s_{10}(x) = \exp(x + 10^{-6})$

**Categorical Features** The dataset includes three categorical features, each with different levels and associated effects:

- **cat_feature_1**: Levels = {A, B, C}. Effects = {A: 0.5, B: -0.5, C: 0.0}
- **cat_feature_2**: Levels = {D, E}. Effects = {D: 1.0, E: -1.0}
- **cat_feature_3**: Levels = {F, G, H, I}. Effects = {F: 0.2, G: -0.2, H: 0.1, I: -0.1}

Each categorical feature is encoded to reflect its specific impact, which varies depending on the level present in the dataset. These effects are designed to simulate real-world scenarios where categorical features may influence the outcome in both positive and negative ways.

Note that the product structure of the considered interaction effects ensures that the true marginal effects $\mathbb{E}[y|x_k]$ are given by $h_k$. This is because, first, with independence of the covariates $x_1, x_2, \ldots x_J$, and $\epsilon$ one has:

$$\mathbb{E}[y|x_k] = \mathbb{E}\left[\sum_{j=1}^{J} s_j(x_j) + \prod x_j + \epsilon \,\middle|\, x_k\right] = s_k(x_k) + \sum_{j=1, j\neq k}^{J} \mathbb{E}[s_j(x_j)] + x_k \prod_{j\neq k} \mathbb{E}[x_j] + \mathbb{E}[\epsilon] \tag{13}$$

Second, assuming zero-centered covariates and effects, as well as a zero-centered error term then yields $\mathbb{E}[y|x_k] = s_k(x_k)$.

### E.1 NETWORK ARCHITECTURES

For the ablation study, fixed network architectures are used and orientated on the literature. Note, that the experiments on real world data are performed with hyperparameter tuning as described in section G. We use NAM architecture inspired by Radenovic et al. (2022) and Dubey et al. (2022). Hence, we use simple network with each feature network consisting of [64, 64, 32] hidden neurons respectively each followed by a 0.1 dropout layer and ReLU activation. For Hi-NAMs we implement the same architecture for the feature interaction network. For EBM and EB$^2$M we use the default architecture (Nori et al., 2019). For GAMs we use cubic splines with 25 knots each. For NAMformer we use an embedding size of 32, 4 layers, 2 heads, 150 one hot encoded bins and dropout of 0.3 throughout all dropout layers, except for a feature dropout of 0.1. Learning rates of 1e-03, a patience of 20 epochs and learning rate decay with a patience of 10 epochs regarding the validation loss was used where applicable.

## F COMPARISON TO FT-TRANSFORMER

We use identical model architecture for both, the NAMformer as well as the FT-Transformer for all datasets. An embedding size of 64, 2 layers, 2 heads, a learning rate of 1e-04, weight decay of 1e-05, task specific head layer sizes of [64, 32], ReLU activation, feed forward dropout of 0.5 and attention dropout of 0.1. For the NAMformer we use feature dropout of 0.1. We use a patience of 15 epochs for early stopping and a learning rate decay with a factor of 0.1 with a patience of 10 epochs with respect to the validation loss.

All datasets are fit using 5-fold cross validation with a validation split of 0.3. Note that as we are implementing 5-fold cross validation we are not using the same splits as for the benchmarks and hence achieve different results.

## G HYPERPARAMETER TUNING

We use Bayesian hyperparameter tuning using the Optuna library (Akiba et al., 2019). We use 50 trials for each method, and report the results for the best trial on either the validation mean squared error or the validation (binary) cross entropy. We use median pruning. For all neural models we implement early stopping with a patience of 15 epochs based on the validation loss and return the best model with respect to the validation loss of that time frame. Additionally we implement learning rate decay with a patience of 10 epochs also based on the validation loss with a factor of 0.1. All hyperparameter spaces for all models are reported below:

**XGBoost**   For the XGBoost model, we tune the following hyperparameters:

- **n_estimators:** Number of trees, varied from 50 to 400.
- **max_depth:** Maximum depth of each tree, with values ranging from 3 to 20.
- **learning_rate:** Learning rate, adjusted between 0.001 and 0.2.
- **subsample:** Subsample ratio of the training instances, from 0.5 to 1.0.
- **colsample_bytree:** Subsample ratio of features for each tree, ranging from 0.5 to 1.0.

**Explainable Boosting Machine (EBM and EB$^2$M)**   For the EBM model, the following hyperparameters are tuned to optimize performance:

- **Interactions:** Set to `0.0` to disable automatic interaction detection.
- **Learning Rate:** The rate at which the model learns, varied from 0.001 to 0.2.
- **Max Leaves:** The maximum number of leaves per tree, with choices ranging from 2 to 4.
- **Min Samples Leaf:** The minimum number of samples required to be at a leaf node, varying from 2 to 20.
- **Max Bins:** The maximum number of bins used for discretizing continuous features, sampled from 512 to 8192.

For EB$^2$M we tune the following:

- **Interactions:** Set to `0.95` to strongly favor interaction effects among features.
- **Learning Rate:** Adjusted identically to the EBM, within the range of 0.001 to 0.2.
- **Min Samples Leaf:** Also ranging from 2 to 20 to control overfitting.
- **Max Leaves:** From 2 to 4, to define the complexity of the learned models.
- **Max Bins:** The binning parameter, ranging from 512 to 8192, to optimize the handling of continuous variables.

**Neural Additive Models (NAMs)**   For Neural Additive Models, we configure a range of hyperparameters to optimize model performance. The hyperparameter space explored includes dimensions of hidden layers, dropout rates, learning rates, and more, as specified below:

- **Hidden Dimensions:** A categorical choice among different layer configurations to adapt the model capacity, including configurations such as `[64, 64]`, `[64, 32, 16]`, `[128, 64]`, `[128, 128, 64]`, `[128, 32]`, and `[128, 64]`.
- **Dropout Rate:** Dropout rate for regularization, varied from 0.1 to 0.5.
- **Feature Dropout Probability:** Probability of dropping a feature to prevent overfitting, ranged from 0.1 to 0.5, sampled on a logarithmic scale.
- **Learning Rate (lr):** Learning rate for training, explored on a logarithmic scale between $10^{-5}$ and $10^{-3}$.
- **Weight Decay:** Regularization parameter to minimize overfitting, also explored on a logarithmic scale, with values ranging from $10^{-6}$ to $10^{-4}$.
- **Activation Function:** The type of activation function used in the model, selected from options such as ReLU, Leaky ReLU, GELU, SELU, and Tanh.
- **Batch size:** Fixed batch size from $\{32, 64, 128, 256, 512\}$.

**Multi-Layer Perceptron (MLP)**   For the MLP model, we fine-tune the following hyperparameters:

- **Hidden Layer Sizes:** Sizes of each hidden layer, where each layer's size is individually tuned between 8 and 512 neurons, defined dynamically for each layer during the trials.
- **Learning Rate (lr):** The optimizer's learning rate, sampled logarithmically between $10^{-5}$ and $10^{-3}$.

- **Weight Decay:** Regularization parameter to minimize overfitting, explored on a logarithmic scale from $10^{-6}$ to $10^{-4}$.
- **Batch Normalization:** A binary choice to either use or not use batch normalization in each layer.
- **Skip Connections:** Option to include skip connections between layers.
- **Activation Function:** Determines the type of activation function used, options include ReLU, Leaky ReLU, GELU, SELU, and Tanh.
- **Dropout Rate:** Dropout rate for each layer to prevent overfitting, adjustable between 0.0 and 0.5.
- **Batch size:** Fixed batch size from $\{32, 64, 128, 256, 512\}$.

**FT-Transformer**    For the FT-Transformer model, we fine-tune the following hyperparameters:

- **Embedding Size:** Dimensionality of embeddings for categorical features, selected from $\{16, 32, 64, 128, 256\}$.
- **Number of Heads (n_head):** The number of attention heads in the transformer, chosen from $\{1, 2, 4, 8\}$.
- **Number of Layers (n_layers):** Configurable number of transformer layers, ranging from 1 to 8.
- **Learning Rate (lr):** Optimized on a logarithmic scale from $10^{-5}$ to $10^{-3}$.
- **Weight Decay:** Regularization parameter explored on a logarithmic scale between $10^{-6}$ and $10^{-4}$.
- **Activation Function:** Options include ReLU, Leaky ReLU, GELU, SELU, and Tanh.
- **Head Dropout:** Dropout rate in the heads, adjustable from 0.0 to 0.5.
- **Attention Dropout:** Dropout rate specifically for the attention mechanisms, also adjustable from 0.0 to 0.5.
- **Head Layer Sizes:** A variety of configurations for layer sizes in the model's head are tested, including:
    - Single layer configurations: $\{1\}$, $\{32\}$, $\{64\}$
    - Dual layer configurations: $\{64, 64\}$, $\{128, 64\}$, $\{128, 32\}$
    - Triple layer configurations: $\{64, 32, 16\}$, $\{128, 128, 64\}$, $\{128, 64, 32\}$
- **Batch Size:** The size of the batches for training, selected from $\{32, 64, 128, 256, 512\}$.

**NAMformer**    For the NAMformer model, we fine-tune the following hyperparameters:

- **Embedding Size:** Dimensionality of embeddings for categorical features, selected from $\{16, 32, 64, 128, 256\}$.
- **Number of Heads (n_head):** The number of attention heads in the transformer, chosen from $\{1, 2, 4, 8\}$.
- **Number of Layers (n_layers):** Configurable number of transformer layers, ranging from 1 to 8.
- **Learning Rate (lr):** Optimized on a logarithmic scale from $10^{-5}$ to $10^{-3}$.
- **Weight Decay:** Regularization parameter explored on a logarithmic scale between $10^{-6}$ and $10^{-4}$.
- **Activation Function:** Options include ReLU, Leaky ReLU, GELU, SELU, and Tanh.
- **Head Dropout:** Dropout rate in the heads, adjustable from 0.0 to 0.5.
- **Attention Dropout:** Dropout rate specifically for the attention mechanisms, also adjustable from 0.0 to 0.5.
- **Feature Dropout:** Dropout rate for features, ranging from 0.1 to 0.5.
    - Single layer configurations: $\{1\}$, $\{32\}$, $\{64\}$

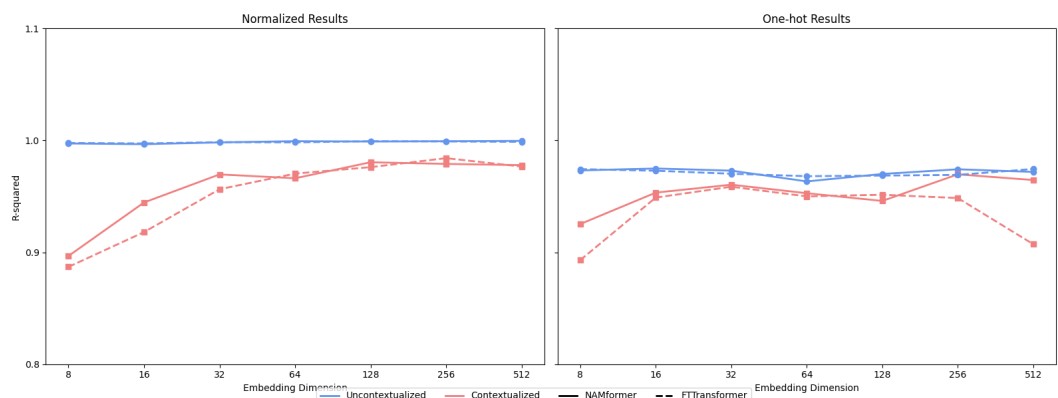

Figure 6: Average $R^2$ values over all 9 features in the california housing dataset. The decision trees are fit, using either the uncontextualized or the contextualized embeddings as training data and the true features as target variables.

- Dual layer configurations: {64, 64}, {128, 64}, {128, 32}
- Triple layer configurations: {64, 32, 16}, {128, 128, 64}, {128, 64, 32}
- **Batch Size:** The size of the batches for training, selected from {32, 64, 128, 256, 512}.

# H   EMBEDDING IDENTIFIABILITY

To demonstrate that the uncontextualized embeddings almost perfectly represent the raw input data, we conducted the experiment described in the main body of our work. We fitted both a NAMformer and a FT-Transformer (Gorishniy et al., 2021) to the California housing dataset, with preprocessing as outlined previously. Our comparison includes both one-hot encoded numerical features and standardized features.

The models were trained using identical architectures with an embedding size of 64, 4 layers, 4 heads, and a uniform dropout rate of 0.3; for the NAMformer, feature dropout was set at 0.1. We employed a learning rate of 1e-04, weight decay of 1e-05, and a patience setting of 15 epochs for early stopping. The reported results represent the average outcomes from a 5-fold cross-validation.

After the models converged, we fitted simple decision trees—using the default settings from (Pedregosa et al., 2011)—to the uncontextualized embeddings from the training split, treating the true features as labels. We then computed the $R^2$ on the test data. Subsequently, we performed the same procedure for the contextualized embeddings. The findings are illustrated in Figure 6.

