# OpenReview forum: "From Uncontextualized Embeddings to Marginal Feature Effects: Incorporating Intelligibility into Tabular Transformer Networks"
_ICLR.cc/2025/Conference — Submitted to ICLR 2025_

### Official Review · Reviewer_FVxa · 2024-11-01

**Soundness:** 2
**Presentation:** 1
**Contribution:** 2
**Rating:** 3
**Confidence:** 2

**Summary:**

This work proposes a modification to one of the tabular transformer architectures that enables features to be interpretable. Specifically, the technique proposed in this work, NAMformer, which is built on top of the existing FT-Transformer, enables the identification of "marginal feature effects" which can be used to interpret the final model. NAMformer performs comparably to the original FT-Transformer, while maintaining desirable interpretability properties.

**Strengths:**

- The claim that the proposed NAMformer is not worse than FT-Transformer in terms of prediction performance seem true on the tasks that were used in the evaluation.
- The general idea of making tabular transformer architectures interpretable seems like it should be useful for a variety of problems.
- The inclusion of synthetic evaluations and ablations are useful for understanding the mechanics of the proposed approach.

**Weaknesses:**

- The introduction and methodology sections could benefit significantly from writing improvements and more thorough explanation. In its current form, these sections assume a lot in terms of prior knowledge, e.g. of marginal feature effects, the FT-Transformer architecture, target-aware embeddings, and uncontextualized embeddings. These should be explained, at least at a high-level. Additionally, from Section 2, it is somewhat unclear which parts are inherited from the prior work on FT-Transformer and which parts are new. Particularly, I was confused about the feature encoding part--without referencing the prior work, I cannot easily assess if this part of the work is novel.
- The authors should include specific use-cases in which the marginal feature effects are useful downstream. Otherwise, it is challenging to assess the significance of the contribution.
- The paper should have a standalone explanation of the datasets that are being used in evaluation beyond saying that the evaluation is done on four regression and four binary classification datasets---what are these datasets and where do they come from? The authors seem to refer the reader to the appendix for this information, but this is critical for assessing the results. Furthermore, the evaluation should be more thorough than regression and binary classification -- conceivably, the proposed method could become worse than FT-Transformer as the number of classes grows. The authors should include such tasks.

**Questions:**

- What are uncontextualized embeddings? As far as I can tell, this is never explained in the paper.
- What does token identifiability mean? This is also not defined.

---

> ### Author Response · Authors · 2024-11-20
> **Answer to Questions and Comments**
>
> Dear reviewer,
> Thank you for your feedback and valuable insights.
>
> Given your comments and questions we were surprised by the particularly low score, especially as several points raised focus on well-established concepts and practices in the field. For example, adjustments to the placement of dataset descriptions (e.g., moving them from the appendix to the main text) are largely structural and do not impact the core contributions or rigor of our approach.
> We have provided detailed responses to each comment below, clarifying how our methods align with robust, widely accepted standards in the literature. We ask that these clarifications be fully considered in your overall assessment of the work.
>
> > The introduction and methodology sections could benefit significantly from writing improvements and more thorough explanation. In its current form, these sections assume a lot in terms of prior knowledge, e.g., of marginal feature effects, the FT-Transformer architecture, target-aware embeddings, and uncontextualized embeddings. These should be explained, at least at a high level.
>
> - Thank you for this suggestion. Our work builds on fundamental concepts from additive modeling ([1, 2, 3, 4]) as well as tabular deep learning ([5, 6]). Following your recommendation, we have added a  “Related Literature” section which is currently placed in the appendix to prevent familiar readers from re-reading well-known concepts. If you feel it would be better suited in the main text or would benefit from additional elements, we are happy to revise accordingly.
>
> > Additionally, from Section 2, it is somewhat unclear which parts are inherited from prior work on FT-Transformer and which parts are new. Particularly, I was confused about the feature encoding part--without referencing the prior work, I cannot easily assess if this part of the work is novel.
>
> - We’re a bit puzzled by this comment. Feature encoding methods are well-established in tabular deep learning, and after double checking we do cite [5] accordingly to clarify that we do not introduce these encodings ourselves.
>
> > The authors should include specific use-cases in which the marginal feature effects are useful downstream. Otherwise, it is challenging to assess the significance of the contribution.
>
> - The usefulness of marginal feature effects is well-established in both statistical modeling and interpretable machine learning ([1, 2, 3, 4, 7, 8]). Marginal feature effects provide direct insight into the relationship between features and model predictions, a cornerstone of interpretability that is critical in domains such as healthcare, finance, and policy-making. These are not niche or isolated use cases but foundational concepts widely recognized for their value in understanding and validating models. The significance of interpretability, particularly through marginal effects, is well-documented in the literature ([1, 2, 3, 4, 7, 8]) and should be evident to any audience, regardless of familiarity with specific methodologies. As such, we believe this comment reflects a misunderstanding of the fundamental role of interpretability in applied machine learning.
>
> - We have added a short paragraph in the "Realted Literature" section in the appendix and could incorporate this into the main part. However, explaining each and every concept of well established themes is beyond a conference submission and boarders on a general and basic introducion into tabular modeling.
>
> > The paper should have a standalone explanation of the datasets that are being used in evaluation beyond saying that the evaluation is done on four regression and four binary classification datasets—what are these datasets and where do they come from?
>
> - We are somewhat surprised by this comment, as the datasets and their sources are described in detail in the appendix, following the standard structure of studies in this field ([1, 2, 3, 4]). Given the page limitations and the general consensus in the field that benchmark dataset descriptions are typically included in the appendix [1–6], we are concerned that moving these descriptions to the main text would compromise the quality of our work. However, we are open to further discussion if you feel strongly otherwise.

---

> ### Author Response · Authors · 2024-11-20
> **Answer Contd.**
>
> > Furthermore, the evaluation should be more thorough than regression and binary classification -- conceivably, the proposed method could become worse than FT-Transformer as the number of classes grows. The authors should include such tasks.
>
> - Please refer to our general answer and our revised manuscript where we have included additional benchmarks. We have oriented on the literature, especially for interpretable modeling, i.e. [1-4, 7-8] and have thus focused on regression and binary classification tasks. It is worth noting, that 90% of datasets in the tabular domain on openML are regression or binary classification task datasets. Since our theoretical justification also holds for multi-class classification (See appendix C) and the number of classes have no effect on the architecture other than the number of output nodes, we disagree that there is any theoretical reasoning why NAMformer should perform worse than FTTransformer for multi-class classification problems.
> To further strengthen this, we have performed 10-fold cross validation on the red and white wine quality datasets taken from [UCI ML Repo](https://archive.ics.uci.edu/dataset/186/wine+quality) and compared NAMformer to FTTransformer with identical hyperparameters. The results show that, as expected, there is no difference for multi-class classification and NAMformer performs just as good as FTTransformer for multi-class classification tasks.
>
> |Model           | Red            | White             |
> |----------------|----------------|-------------------|
> | NAMformer      | 0.607  ± 0.035 | 0.569 ± 0.011     |
> | FTTransformer  | 0.603  ± 0.035 | 0.551  ± 0.015    |
>
>
>
> # Questions
> > What are uncontextualized embeddings
> - Uncontextualized embeddings are fixed vector representations of words/inputs/tokens, generated independently of the surrounding context. That is, the embeddings before being passed through any joint layers.
>
> > What does token identifiability mean
> - Token identifiability refers to the ability to distinguish a specific token (e.g., a word/input) within a sequence/features based on its unique representation. It ensures that the model can differentiate tokens, even in similar or repeated contexts, to preserve meaning and structure. Please refer to [9] for a detailed description.
>
> Both, token identifiability and (un)contextualized embeddings are well known concepts in ML, which is why we did not include specific definitions in our paper.
>
> ---
>
> Thank you once again for your feedback. We hope these responses provide clarification and reinforce our contributions.
>
>
> ---
> [1] Agarwal, Rishabh, et al. "Neural additive models: Interpretable machine learning with neural nets." NeurIPS 2021
> [2] Chang, Chun-Hao,et al. "NODE-GAM: Neural Generalized Additive Model for Interpretable Deep Learning." ICLR 2022.
> [3] Thielmann, Anton, et al. "Interpretable Additive Tabular Transformer Networks." TMLR 2024.
> [4] Radenovic, Filip, et al. "Neural basis models for interpretability." NeurIPS 2022.
> [5] Gorishniy, Yury, et al. "On embeddings for numerical features in tabular deep learning." NeurIPS 2022.
> [6] Gorishniy, Yury, et al. "Revisiting deep learning models for tabular data." NeurIPS 2021.
> [7] Hastie, Trevor J. "Generalized additive models." Statistical models in S." Routledge, 2017.
> [8] Hastie, Trevor, and Robert Tibshirani. "Generalized additive models: some applications." JASA 1987.
> [9] Brunner, Gino, et al. "On identifiability in transformers." arXiv preprint arXiv:1908.04211 (2019).

---

> > ### Author Response · Authors · 2024-11-29
> > **Invitation to discussion**
> >
> > Dear Reviewer,
> >
> > We have done our best to address the concerns expressed in your review. Specifically, we:
> >
> > - Performed additional benchmarks to verify the superiority of NAMformer compared to other interpretable models.
> > - Added a section on related literature.
> > - Demonstrated that NAMformer can match the performance of FTTransformer on multi-class classification problems.
> > - Addressed all of your other questions.
> >
> > We look forward to hearing your feedback and continuing the discussion.

---

### Official Review · Reviewer_hMeM · 2024-11-02

**Soundness:** 2
**Presentation:** 2
**Contribution:** 2
**Rating:** 5
**Confidence:** 2

**Summary:**

This work proposed an adaptation of tabular transformer networks designed to identify marginal feature effects, while maintaining the capability of prediction performance at the original level.

**Strengths:**

From my prospective, the strengths of this paper are as follows:
- Compared to FT-Transformer or GAM, the proposed NAMformer has more interpretability and stronger predictive power, respectively
- There are analysis between contextual and uncontextual embeddings, proving that the uncontextual embedding can represent the original feature
- There are a lot of formula providing theoretical support

**Weaknesses:**

After reading the paper, I still have some questions.
1. In section 2, the encoding of numerical features involves the threshold $b_t$, and the thresholds are from the decision trees. So I wonder how we get these decision trees? Is the marginal effects of the model the same as those decision trees? And what about the performance compared with the decision trees.
2. The method seems too trivial to achieve the expected performance. Only a Linear and dropout can identify the marginal and even enhance the performance.
3. In tabular domain, it seems that 8 datasets in all are not sufficient. This may result in doubt that the datasets used are selected for this task.

In addition to the above questions, I think there are some problems with typography and content。
1. It will be better if there is a section which introduces the related work, e.g. transformers in tabualr, marginal feature effects in tabular or other domains, etc. I think this will help reader better understand the task and your method.
2. It seems that in section Abstract, there is some problem with the spacing between lines.

**Questions:**

See Weaknesses

---

> ### Author Response · Authors · 2024-11-20
> **Answer to Questions and Weaknesses**
>
> Dear reviewer,
> Thank you for your valuable feedback and insights. We answer your questions and comments below.
>
> ## Weaknesses
>
> > In section 2, the encoding of numerical features involves the threshold $b_t$, and the thresholds are from the decision trees. So I wonder how we get these decision trees? Is the marginal effect of the model the same as those decision trees? And what about the performance compared with the decision trees?
>
> Thank you for highlighting this point. Firstly, we acknowledge a typo in the PLE encoding. The correct form should be as follows:
>
> \\[
> z_{j(\text{num})}^{t} =
> \begin{cases}
> 0, & \\
> \\text{if } x < b_{t-1} \\\\
> 1, & \\text{if } x \geq b_t \\\\
> \\frac{x - b_{t-1}}{b_{t} - b_{t-1}}, & \\text{otherwise}
> \\end{cases}
> \\]
>
> We have corrected it in the revised version of the manuscript.
>
> In our approach, the encodings are computed in a target-aware manner. Specifically, as a preprocessing step, we fit a decision tree on pairs of the form $(x_j, y)$ for each feature \( j = 1, ..., J \). The bins derived from the decision tree then define $b_t$. However, it’s important to note that the true value of $x$ is retained in the encoding, resulting  $x \\mapsto z $, where $z \\in \\mathbb{R}^T$ is an enriched, higher-dimensional representation of $x$. Consequently, each feature subnetwork $f_j $ can yield different marginal predictions compared to the decision tree. Please also refer to [1] for a detailed introduction into embeddings for numerical features.
>
> > The method seems too trivial to achieve the expected performance. Only a linear layer and dropout can identify the marginal effects and even enhance the performance.
>
> - Thank you for acknowledging the simplicity of our approach. A useful analogy is the `mgcv::gam` package in R, which transforms each feature into a higher-dimensional representation. In the case of GAMs, this is achieved through splines, whereas in our approach, we use PLE followed by the embedding layer. Splines are well-known for their ability to effectively capture marginal effects, as supported by sources [2, 3, 4].
> As such, each feature network practically consists of an embedding layer, a possible activation function and an output layer. Since the input is already transformed to be in  $z \\in \\mathbb{R}^T$, we find that these networks are capable of identifying marginal effects well as demonstrated in Table 1 and the provided figures.
>
> > In the tabular domain, it seems that only 8 datasets are used, which may raise doubts that these datasets were selected for this task.
>
> - Thank you for this comment. We have based our selection on commonly used datasets in additive modeling. These datasets are also employed in studies such as [4, 5, 6, 7] and exceed the number used in those references. Nonetheless, as part of this revision, we have added additional datasets to the benchmark. So far, we have added 7 more datasets for 7 models, with ongoing efforts to expand both datasets and model comparisons. Preliminary results with these additional datasets align closely with the outcomes reported in our paper. Please see our general answer for the results.
>
> > It would be helpful to include a section on related work, such as transformers in tabular data, marginal feature effects in tabular or other domains, etc. This could help readers better understand the task and your method.
>
>  - Thank you for your valuable suggestion. We have included a section on related work, discussing transformers for tabular data and marginal feature effects, to provide additional context for readers. Currently, we have placed it in the appendix to prevent familiar readers from re-reading well-known concepts. If you feel it would be better suited in the main text or would benefit from additional elements, we are happy to revise accordingly.
>
> > It seems that in the Abstract, there are some spacing issues between lines.
>
> Thank you for noticing this. We have adapted it in our revised version.
>
> ---
>
> Thank you once again for your thoughtful feedback. We hope these responses provide clarification and reinforce our contributions.
>
> ---
> [1] Gorishniy, Yury, et al. "On embeddings for numerical features in tabular deep learning." NeurIPS. 2022
> [2] Rügamer, David, et al. "Semi-structured distributional regression." The American Statistician. 2024
> [3] Wood, Simon N. "Thin plate regression splines." Journal of the Royal Statistical Society Series B. 2003
> [4] Perperoglou, Aris, et al. "A review of spline function procedures in R." BMC medical research methodology. 2019

---

> > ### Author Response · Authors · 2024-11-20
> > **Answer Contd.**
> >
> > > In practice, how can we be confident that the feature networks have the capacity to fully capture marginal effects? If they can't fully capture them, then part of the marginal effect could be learned by the transformer instead.
> > - In practice, we have observed that effects identifiable by models such as NAM or GAM can also be effectively captured—if not improved upon—by the NAMformer (as demonstrated in Table 1). Single-feature effects are typically not so complex that a well-tuned neural network cannot model them adequately, and our experiments did not reveal any limitations in this regard. On the contrary, we found that, due to the smaller model size of the single-feature networks, the model identifies marginal feature effects even with very small values of feature dropout, i.e., without any form of imposed identifiability constraint. This is presumably due to the fact, that the smaller feature networks converge relatively faster, than the backbone FT-Transformer architecture and thus ensure proper identification of marginal effects. However, you are correct that, as with all neural models (e.g., NAMs [1], NodeGAM [2], NAMLSS [3], NBM [4]), it is inherently challenging to ensure absolute certainty about the significance of the identified effects.
> >
> > ---
> >
> > Thank you once again for your feedback. We hope these responses provide clarification and reinforce our contributions.
> >
> > ---
> >
> > [1] Agarwal, Rishabh, et al. "Neural additive models: Interpretable machine learning with neural nets." NeurIPS 2021
> > [2] Chang, Chun-Hao,et al. "NODE-GAM: Neural Generalized Additive Model for Interpretable Deep Learning." ICLR 2022.
> > [3] Thielmann, Anton, et al. "Neural Additive Models for Location Scale and Shape." AISTATS 2024.
> > [4] Radenovic, Filip, et al. "Neural basis models for interpretability." NeurIPS 2022.

---

> > ### Comment · Reviewer_hMeM · 2024-11-25
> >
> > Thank you for the clarifications.
> >
> > From my perspective, the related work should be included in the main body of the paper, as this would enhance the completeness and readability of your paper (this will not affect my evaluation). Regarding the $b_t$, since you are fitting a decision tree for each feature, can I interpret this as leveraging the power of decision trees to drive the performance, rather than attributing the effect to your structural design? I would appreciate a clearer distinction between your method and decision trees.
> >
> > Thank you for the explanation of my question, but I am still sceptical that one MLP layer can achieve performance in the paper, and I will refer to the paper you recpmmend for a more comprehensive understanding.

---

> ### Author Response · Authors · 2024-11-25
> **Answer II**
>
> Dear Reviewer,
>
> Thank you for your detailed feedback. We appreciate your time and suggestions.
>
> > From my perspective, the related work should be included in the main body of the paper, as this would enhance the completeness and readability of your paper.
>
> - Thank you for your valuable suggestion. We will include the related work in the main body of the final version of the paper to enhance its completeness and readability.
>
> > Regarding the $b_t$, since you are fitting a decision tree for each feature, can I interpret this as leveraging the power of decision trees to drive the performance, rather than attributing the effect to your structural design? I would appreciate a clearer distinction between your method and decision trees.
>
> - The PLE encoding, proposed by [1], is strictly a preprocessing method which can also be left out. Your interpretation is quite correct, in that we leverage it purely for performance reasons (see Table 2).
> We do not utilize predictions or any decision tree outputs other than the bin boundaries, $b\_t$. These boundaries can also be set continuously or based on data occurrence without relying on decision trees. For instance, methods like splines often use such continuous boundaries to great effect (see e.g. [2, 3, 4]).
>
> - To further clarify, Table 1 illustrates that our method works effectively with alternative encodings, such as one-hot encoding or standardized input features, independent of decision trees. This versatility is further demonstrated in Table 2, where alternative encodings are also evaluated, showing that while decision-tree-derived boundaries offer a slight performance advantage, they are not a structural requirement of our method.
>
>
> > I am still skeptical that one MLP layer can achieve the performance claimed in the paper, and I will refer to the paper you recommend for a more comprehensive understanding.
>
> - While we greatly value your skepticism in the review process, our claims are well-supported by extensive literature, experimental validation and our provided [Repository](https://anonymous.4open.science/r/nmfrmr-F76D/README.md). Single-layer architectures have demonstrated great performance, especially in Neural Additive Models (NAMs) and similar frameworks. For example, [4] shows that a single-layer, spline-based NAM outperforms more complex NAMs and Explainable Boosting Machines (EBMs). Similarly, [5] demonstrates the efficacy of shallow, parameter-efficient networks using low-rank polynomials.
>
> - Furthermore, our study focuses on analyzing marginal effects without involving complex higher-order feature interactions, which aligns with the suitability of a shallow architecture. We encourage you to explore the provided codebase for additional empirical evidence. Given the strong support and reproducibility demonstrated in our approach, we hope this provides clarity and resolves your skepticism.
>
>
> ---
>
> > In the tabular domain, it seems that eight datasets are not sufficient. This may result in doubt that the datasets used are selected for this task.
>
> - In our previous response, we addressed your concern regarding the number of datasets by nearly doubling the benchmarks, which required substantial effort. Since you had highlighted limited experiments as a core weakness, we kindly request that this significant improvement be acknowledged in your evaluation.
>
> ---
>
> Thank you once again for your insightful feedback. We are committed to further improving our work based on your comments.
>
>
> ---
> [1] Gorishniy, Yury, et al. "On embeddings for numerical features in tabular deep learning." NeurIPS. 2022.
> [2] Wood, Simon N. "Generalized additive models: an introduction with R." (2017).
> [3] Yeh, Raine, et al. "Fast automatic knot placement method for accurate B-spline curve fitting." Computer-aided design 128 (2020).
> [4] Luber, M., et al. "Structural neural additive models: Enhanced interpretable machine learning." arXiv preprint (2023).
> [5] Dubey, A., et al. "Scalable interpretability via polynomials." NeurIPS (2022).

---

> ### Author Response · Authors · 2024-11-29
> **Invitation to continue the discussion**
>
> Dear reviewer, we did our best to address your remaining concerns and answer your questions. In summary, we have:
>
> - Detailed that decision trees are exclusively used during preprocessing and are only leveraged to determine the bin boundaries $b\_t$ and nothing else.
> - Performed additional benchmarks to verify the superiority of NAMformer compared to other interpretable models.
> - Addressed your remaining questions.
>
> We look forward to hearing your feedback and continuing the discussion.

---

### Official Review · Reviewer_2nBF · 2024-11-04

**Soundness:** 3
**Presentation:** 3
**Contribution:** 4
**Rating:** 8
**Confidence:** 4

**Summary:**

This paper proposes a modification to predictive tabular transformer models to make them more interpretable: producing the final output as a sum of the transformer output along with shallow univariate models for each input feature. Dropout is used to encourage the model to optimize its use of each feature independently. The univariate models can then be interpreted as indicating the marginal effect of each feature on the overall output.

**Strengths:**

- In general, the proposed method makes sense to me and I could see it being used in practice. I've worked with tabular transformers in applied settings and can attest that understandable per-feature explanations are difficult to achieve. This method makes sense to me as a way to get greater interpretability while retaining the performance of tabular transformers, or other deep tabular models.
- As far as I can tell the proposed method is original. It resembles NATT (Thielmann et al. 2024), but I think the design differences are significant in practice and make sense. This method passes all features to the transformer as input, then uses dropout during training to guide the model to a desired behaviour that makes all features interpretable, whereas NATT keeps numeric and categorical features separate and only provides a comparable level of interpretability for numeric features.
- Experimental results are fairly convincing in showing that the proposed method doesn't sacrifice performance.

**Weaknesses:**

- Discussion of related works is lacking. There is no separate related works section, just some context provided in the introduction, and while a decent selection of related papers are cited, there's little discussion of how the proposed method relates to and differs from existing work. One point I would like to see addressed is whether and how this method improves on methods like integrated gradients, LIME, or SHAP that produce per-feature explanations without requiring model modifications.

- Some theoretical results are given in Section 2.1, but it's not described how the results relate back to the interpretability of the model, and there are some particular issues I wanted to raise:
	- The definition given for $R_{\tilde{w}_{-k}}$ at line number 279 doesn't make sense to me - it could even take the opposite sign of the overall risk \(R\) if the model has a greater loss with just feature $k$ than on average.
	- Equation (7) depends on the assumption that the risk for each single-feature dropout vector is the same and equal to $R(1-p(\tilde{w}_k))$, which seems very unlikely for realistic models or datasets where features have different importances and the model tends to perform better given more features. So I don't see the relevance of this result.
	- My interpretation of the section is that when training with dropout, the performance of individual feature predictors is in some sense bounded by the overall model performance. This makes sense, but it's not clear to me if this indicates anything about the interpretability of the model in terms of the individual feature predictors or what their predictions represent (i.e., do they actually optimize the loss with respect to $\mathbb{E}_{y|x_k}[y|x_k]$, or just something that's not too far from it?).

- Even under the proposed approach, the transformer component itself is somewhat of a black box. This could be tolerable given the improved interpretability of the overall model, but it is a limitation.
	- While thinking through the method, I kept coming back to what the optimal behaviour learned by the transformer and shallow predictors would be. I think it is true that the optimal behaviour at low dropout rates is at least approximately for the transformer to only predict joint effects plus marginal effects that are too complex for the shallow predictors to learn, rather than learning all joint and marginal effects. A proof of this would be useful if possible though.

Given that I think the proposed method could have real practical utility, I'm leaning towards an accept, but I would prefer to see more a more relevant theoretical discussion.

**Questions:**

- Could you please clarify the results in Section 2.1, especially how to interpret them with respect to interpretability?
- In practice, how can we be confident that the feature networks have the capacity to fully capture marginal effects? If they can't fully capture them, then part of the marginal effect could be learned by the transformer instead.

---

> ### Author Response · Authors · 2024-11-20
> **Answer to Comments and Questions**
>
> Dear reviewer,
> Thank you for your valuable feedback and insights.
>
>
> > The definition given for at line number 279 doesn't make sense to me - it could even take the opposite sign of the overall risk $R\_{\\tilde{\\mathbf{w}}\_{-k}}$ if the model has a greater loss with just feature than on average.
>
> - We fully agree that the describing $R\_{\\tilde{\\mathbf{w}}\_{-k}}$ as the "risk not associated with $\\tilde{\\mathbf{w}}\_{-k}$" is misleading. We think that the term "difference between the overall risk and the risk associated with $\\tilde{\\mathbf{w}}\_{-k}$" would be more appropriate. We have revised the manuscript accordingly. We also agree that this quantity is negative if the model has a greater loss with just feature $k$ than on average, which will typically be the case.
>
> This further makes sense in the context of equation (5) as the risk regarding the marginal effect $\\mathbb{E}\_{x\_k, y \\sim P^{\\mathcal{D}}} \\left [\\mathcal{L} \\left ( \\beta\_0 + f\_k(x\_k) , y\\right)\\right ]$ is decreased by a negative quantity $R\_{\\tilde{\\mathbf{w}}\_{-k}}(1-p(\\tilde{\\mathbf{w}}\_k))$ compared to the overall risk $R$. I.e., $- R\_{\\tilde{\\mathbf{w}}\_{-k}}(1-p(\\tilde{\\mathbf{w}}\_k))$ increases the risk of identifying the marginal effect.
>
> $$\\mathbb{E}\_{x\_k, y \\sim P^{\\mathcal{D}}} \\left [\\mathcal{L} \\left ( \\beta\_0 + f\_k(x\_k) , y\\right)\\right ]  =  \\frac{R - R\_{\\tilde{\\mathbf{w}}\_{-k}}(1-p(\\tilde{\\mathbf{w}}\_k))}{p(\\tilde{\\mathbf{w}}\_k)}  \\ \\ (5).$$
>
> > Equation (7) depends on the assumption that the risk for each single-feature dropout vector is the same and equal to
> $R\_{\\tilde{\\mathbf{w}}\_{-k}} = R \\cdot (1-p(\\tilde{\\mathbf{w}}\_k)) $, which seems very unlikely for realistic models or datasets where features have different importances and the model tends to perform better given more features. So I don't see the relevance of this result.
>
> - We included this result because it provides an intuitive upper bound on the risk for identifying a specific marginal effect $\\mathbb{E}\_{y|x\_k}{\\left [ y | x\_k  \\right ] }$. We agree that the assumption of equal risk for each single-feature dropout vector is strong and may not hold in practice. However, we think that this result is still useful for understanding the behavior of the model and the dropout procedure. We now clearly point out in the manuscript that this case is unrealistic and that the result is a theoretical upper bound.
>
> > Could you please clarify the results in Section 2.1, especially how to interpret them with respect to interpretability?
>
> - Our goal of section 2.1 is to relate the risk in identifying a marginal effect, i.e. $\\mathbb{E}\_{x\_k} \\left [ \\mathcal{L}\\left ( \\beta\_0 + f\_k(x\_k), \\mathbb{E}\_{y|x\_k}{\\left [ y | x\_k  \\right ] } \\right ) \\right ]$ to the overall risk $R$ when performing feature dropout in order to provide theoretical insights. The finding in equation (6) can be summarized as having proven that minimizing the ratio between the difference between overall risk and the risk associated with $\\tilde{\\mathbf{w}}\_{-k}$ and, second, the dropout probability for only keeping the $k$-th vector implies a low risk in identifying marginal effects. We added this explanation to the manuscript to clarify the interpretation of the results.
>
> - While stricter theoretical bounds would certainly be ideal, even though perhaps not even possible, we believe that our results and framework still provide valuable insights into the behavior of feature dropout. Especially because, to our very best knowledge, we are first to provide such a theoretical analysis.
>
> - We would also like to point out that we provide additional comprehensive empirical results in section 3 that show that NAMFormer outperforms other commonly used methods in terms of identifiability of marginal effects---even GAMs and linear models. It is thus very likely that, for instance the fact that the NAM-Part of our methodology can fit the marginal effects of the model very well, is a key factor in the success of our method which we do not capture in the theoretical analysis.

---

> > ### Author Response · Authors · 2024-11-20
> > **Answer Continued**
> >
> > > In practice, how can we be confident that the feature networks have the capacity to fully capture marginal effects? If they can't fully capture them, then part of the marginal effect could be learned by the transformer instead.
> > - In practice, we have observed that effects identifiable by models such as NAM or GAM can also be effectively captured—if not improved upon—by the NAMformer (as demonstrated in Table 1). Single-feature effects are typically not so complex that a well-tuned neural network cannot model them adequately, and our experiments did not reveal any limitations in this regard. On the contrary, we found that, due to the smaller model size of the single-feature networks, the model identifies marginal feature effects even with very small values of feature dropout, i.e., without any form of imposed identifiability constraint. This is presumably due to the fact, that the smaller feature networks converge relatively faster, than the backbone FT-Transformer architecture and thus ensure proper identification of marginal effects. However, you are correct that, as with all neural models (e.g., NAMs [1], NodeGAM [2], NAMLSS [3], NBM [4]), it is inherently challenging to ensure absolute certainty about the significance of the identified effects.
> >
> > ---
> >
> > Thank you once again for your feedback. We hope these responses provide clarification and reinforce our contributions.
> >
> > ---
> >
> > [1] Agarwal, Rishabh, et al. "Neural additive models: Interpretable machine learning with neural nets." NeurIPS 2021
> > [2] Chang, Chun-Hao,et al. "NODE-GAM: Neural Generalized Additive Model for Interpretable Deep Learning." ICLR 2022.
> > [3] Thielmann, Anton, et al. "Neural Additive Models for Location Scale and Shape." AISTATS 2024.
> > [4] Radenovic, Filip, et al. "Neural basis models for interpretability." NeurIPS 2022.

---

> > > ### Comment · Reviewer_2nBF · 2024-11-22
> > >
> > > Thank you for the improvements and corrections.
> > >
> > > The added literature review is useful - I would definitely suggest adding at least some of it to the main body so that readers are more aware of the context of the work. I still would have preferred to see an explicit comparison of the proposed method to post-hoc feature attribution methods like integrated gradients, because I think a natural question when looking at this work is "why not just train an ordinary FT-Transformer and estimate feature effects afterwards?".
> > >
> > > > We agree that the assumption of equal risk for each single-feature dropout vector is strong and may not hold in practice. However, we think that this result is still useful for understanding the behavior of the model and the dropout procedure.
> > >
> > > Could you please elaborate on how this is useful? From what I can tell, the bound is only that tight under ideal scenarios, so I don't see what this adds in comparison to (7).
> > >
> > > I feel better about the correctness of the theoretical analysis after the changes, but I'm still doubtful about its utility. I'll go through Appendix C and take a closer look at the simulation study.

---

> ### Author Response · Authors · 2024-11-23
> **Response II**
>
> Dear reviewer,
>
> Thank you for your detailed follow-up and for carefully reviewing the updates we have provided.
>
>
> > The added literature review is useful - I would definitely suggest adding at least some of it to the main body so that readers are more aware of the context of the work.
> - Thank you for  this advice. We will await the other reviewers comments before adapting these changes, but we will definitely include a broader literature review in our final version.
>
>
> > I still would have preferred to see an explicit comparison of the proposed method to post-hoc feature attribution methods like integrated gradients, because I think a natural question when looking at this work is "why not just train an ordinary FT-Transformer and estimate feature effects afterwards?".
>
> - Thank you for this question. While Integrated Gradients (IG) [1] and other post-hoc methods are powerful for detecting feature importance, particularly in classification tasks, they may face challenges in identifying global marginal effects, especially for regression tasks. Some of these challenges and limitations are discussed in [2].
> For IG in particular.
>
> - **Baseline Sensitivity and Path Dependence**:
>    IG computes attributions by integrating gradients along a path from a baseline $\\mathbf{x}\_{\\text{baseline}}$ to the input $\\mathbf{x}$:
>    $
>    \\text{IG}\_i(\\mathbf{x}) = (x\_i - x\_{i, \\text{baseline}}) \\int\_{\\alpha=0}^1 \\frac{\\partial f(\\mathbf{x}')}{\\partial x\_i} \\Big|\_{\\mathbf{x}' = \\mathbf{x}\_{\\text{baseline}} + \\alpha(\\mathbf{x} - \\mathbf{x}\_{\\text{baseline}})} d\\alpha.
>    $
>    The choice of baseline affects how the path traverses the input space, influencing attributions significantly, especially for features involved in interactions. For instance, the attribution of a feature $x\_i$ in the presence of an interaction term $f\_{12}(x\_i, x\_j)$ depends on the values of $x\_j$ along the path. A poorly chosen baseline may exaggerate or understate the contribution of interactions, leading to attributions that are inconsistent with the actual marginal effect.
>
>
> - To illustrate this, we have added a Jupyter Notebook in our [Github repository](https://anonymous.4open.science/r/nmfrmr-F76D/integrated_gradients.ipynb). The notebook demonstrates that IG fails to accurately identify marginal effects, even in cases of simple additive data-generating processes without interaction effects.
>
> Additionally, we included an FT-Transformer in our ablation study and extracted marginal effects using Integrated Gradients, as per your suggestion. The results, shown below, demonstrate that IG struggled to accurately identify marginal effects in regression problems with feature interactions, consistently performing poorly and even yielding negative correlations across the board.
> While these results at a first glance might be surprising, the results in our notebook (plots are at the bottom) demonstrate the methods struggles, even for datasets without any interactions.
>
>
> | Model                 | 3            | 4            | 5            | 6            | 7            | 8            | 9            |
> |-----------------------|--------------|--------------|--------------|--------------|--------------|--------------|--------------|
> | Linear                | 0.467 ±0.41  | 0.251 ±0.67  | 0.220 ±0.62  | 0.238 ±0.59  | 0.124 ±0.63  | 0.034 ±0.68  | 0.092 ±0.68  |
> | GAM                   | 0.800 ±0.37  | 0.534 ±0.76  | 0.466 ±0.73  | 0.500 ±0.69  | 0.356 ±0.76  | 0.257 ±0.81  | 0.299 ±0.79  |
> | NAMformer             | 0.806 ±0.40  | 0.918 ±0.15  | 0.879 ±0.17  | 0.867 ±0.11  | 0.909 ±0.10  | 0.617 ±0.59  | 0.756 ±0.56  |
> | FTTransformer (IG)    | -0.117 ±0.25 | -0.120 ±0.22 | -0.20 ±0.311 | -0.301 ±0.37 | -0.332 ±0.41 | -0.380 ±0.42 | -0.440 ±0.41 |

---

> > ### Author Response · Authors · 2024-11-23
> > **Response II Contd.**
> >
> > > Could you please elaborate on how this is useful? From what I can tell, the bound is only that tight under ideal scenarios, so I don't see what this adds in comparison to (7).
> >
> > - Thank you for your thoughtful question. We agree that the assumption of equal risk for each single-feature dropout vector is idealized and that the bound is tight primarily under such scenarios. However, we believe this result remains useful for several reasons:
> >
> >   1. **Theoretical Insight into Dropout Behavior**:
> >    The derived bound highlights the relationship between the risk of identifying marginal effects, dropout probabilities, and overall risk. While simplified, it offers an **intuitive understanding** of how feature dropout interacts with model behavior.
> >
> >   2. **Practical Relevance**:
> >    Despite the idealized assumption, our empirical results (Section 3) show that the proposed methodology consistently outperforms methods like GAMs and linear models in identifying marginal effects. This suggests that the theoretical framework, while simplified, captures key factors underlying its success in practice.
> >
> >   3. **Foundation for Future Work**:
> >    By explicitly analyzing the equal-risk assumption, this result lays a foundation for refining these bounds under more realistic conditions. It provides a starting point for understanding dropout’s role in marginal effect identifiability.
> >
> > - However, we do not want to create the impression of overselling our work or presenting misleading results. Hence, we are happy to adjust any wordings or formulations that may appear unclear or overstated.
> >
> > > I feel better about the correctness of the theoretical analysis after the changes, but I'm still doubtful about its utility. I'll go through Appendix C and take a closer look at the simulation study.
> > - Thank you for acknowledging our changes. We encourage you to revisit the ablation study (section 3) and the included effect plots in our manuscript, which we believe effectively demonstrate the utility of our model.
> >
> >
> > ---
> >
> > Thank you once again for your thoughtful engagement. We hope our responses address your concerns and would be glad to continue the discussion further if needed.
> >
> > ---
> >
> > [1] Sundararajan, M., et al. "Axiomatic attribution for deep networks." ICML.  2017
> > [2] Molnar, C., et al. "General pitfalls of model-agnostic interpretation methods for machine learning models." Springer International Publishing, 2020.

---

> > > ### Author Response · Authors · 2024-11-29
> > > **Invitation to continue the discussion**
> > >
> > > Dear reviewer, we did our best to address your remaining concerns and answer your questions. In summary, we have:
> > >
> > > - Distinguished NAMformer from Integrated Gradients and including your suggestion of fitting an FTTransformer and analyzing the marginal effects via integrated gradients.
> > > - Answered your question regarding the equal risk assumption
> > >
> > > We would greatly appreciate it if you could let us know if you have any remaining concerns. If not, we kindly ask you to consider reflecting these changes in your scores.
> > >
> > > We look forward to your feedback and to continuing the discussion.

---

> > > > ### Comment · Reviewer_2nBF · 2024-12-03
> > > >
> > > > Thank you for the discussion. Since the main issues I raised with the content have been mostly addressed, I've raised my score. I would note that this is based on the expectation that the authors will move their related work section into the main body of the paper, but they have already replied to reviewer hMeM that they would do so, and there's enough room in the main body for it to fit.
> > > >
> > > > My positive assessment is still mainly coming from my view as a practitioner that I could see this method being used in real-world settings. A key part of validating and monitoring tabular models in high-risk domains is assessing feature contributions, often in terms of global trends for how each feature individually affects outputs. This method provides those in a way that strikes me as more verifiable and understandable than the post-hoc attribution methods that are available for tabular deep learning models. I think the experiments provided are adequate for assessing the key question of whether the proposed modifications hurt predictive performance, and it appears that they generally don't. I would also note that the architectural changes would easily translate to other supervised tabular neural network models than FT-Transformer, which is useful given the speed that models are changing at in this field.
> > > >
> > > > Weaknesses I still see with the paper are that the theoretical analysis still seems limited in significance to me, and there are some interpretability challenges with the method that seem tough to avoid, like the transformer still being a black box and possibly picking up some marginal effects itself. However, the experimental results provide some evidence that these aren't major issues and in practice, I think these drawbacks could be acceptable tradeoffs when opting for complex tabular NN models over simpler models.

---

> ### Author Response · Authors · 2024-12-03
> **Closing Comment: Thank You for Your Engagement**
>
> Dear Reviewer,
>
> Thank you for your thoughtful feedback and continued support. We’re happy to hear that your concerns have been addressed.
> We will move the related work section into the main body in the final version as suggested. We also plan to expand the discussion and limitations section to better incorporate your comments, building on the points already addressed.
>
> ---
> Thank you again for your insights and for raising your score. We’re committed to making further improvements and ensuring the manuscript reflects your helpful suggestions.

---

### Official Review · Reviewer_2Ug9 · 2024-11-04

**Soundness:** 2
**Presentation:** 2
**Contribution:** 2
**Rating:** 5
**Confidence:** 1

**Summary:**

I do not feel qualified to review this paper as I do not have any experience with tabular machine learning. I would like to ask the ACs to seek an opinion from different reviewers.

**Strengths:**

n/a

**Weaknesses:**

n/a

**Questions:**

n/a

---

### Author Response · Authors · 2024-11-20
**General Response to all reviewers**

Dear Reviewers,

We would like to sincerely thank you for taking the time to review our work and providing thoughtful feedback. We greatly appreciate your evaluations, which we find encouraging as they underline the key benefits of our proposed approach.

**Edit**: Since the Github link in the provided PDF expired, we have created a new link, including the same repository as before: [Repository](https://anonymous.4open.science/r/nmfrmr-F76D/README.md)

We are confident that we can effectively address the concerns you raised and have already taken steps to improve the manuscript. Specifically:

- **Dedicated Literature Review:** To provide better context for readers unfamiliar with Tabular Deep Learning and Additive Models, we have included a focused literature review explaining these concepts in greater detail. This section is currently placed in the appendix, as we aim to direct unfamiliar readers to that section without disrupting familiar readers with content they are likely already acquainted with. However, we are happy to integrate it into the main body of the paper after further discussion and realignment with you.

- **Additional Benchmarks:** To further strengthen the claims of our study, we have conducted additional benchmarks. Given the short time frame and limited computational resources, we focused on regression tasks and interpretable models, standard normalized the response variables and applied the same hyperparameter optimization as in the manuscript. We are currently conducting further benchmarks and will include these additional results in the final version.
The results from these additional benchmarks, performed on seven new datasets from OpenML (ID 353; [1]), strongly support the conclusions presented in the original manuscript. These findings reinforce the robustness and reliability of our approach.

We hope these updates demonstrate our commitment to addressing your concerns and improving the clarity and rigor of our work. We are confident that the revised manuscript presents a more compelling and comprehensive case for our contributions. Additionally, we have adressed your individual questions and concerns in dedicated answers below.


Thank you again for your constructive feedback, and we look forward to your further input.

**Additional Benchmark results for interpretable models:**

| Model       | AV ↓    | GS ↓   | K8 ↓   | P32 ↓  | MH ↓   | BH ↓   | SG ↓   |
|-------------|---------|--------|--------|--------|--------|--------|--------|
| Linear      | 0.700   | 0.366  | 0.580  | 0.843  | 0.295  | 0.025  | 0.445  |
| GAM         | 0.287   | 0.228  | 0.557  | 0.909  | 0.157  | 0.023  | 0.273  |
| EBM         | 0.050   | 0.079  | 0.411  | 0.395  | 0.096  | 0.033  | 0.272  |
| EB²M        | 0.049   | 0.079  | 0.409  | 0.388  | **0.099** | 0.026  | **0.263** |
| NAM         | 0.372   | 0.235  | 0.927  | 1.049  | 0.181  | 0.025  | 0.399  |
| Hi-NAM      | 0.135   | **0.034** | **0.076** | 0.435  | 0.102  | 0.128  | 0.278  |
| NAMformer   | **0.023** | 0.051  | 0.108  | 0.397  | **0.095** | **0.022** | 0.270  |


Average performance over all 15 datasets, i.e. the datasets from the original submission as well as the additional datasets:

| Model       | Average Rank | Rank Std Dev |
|-------------|--------------|--------------|
| NAMformer   | **1.700**    | 0.678        |
| EB²M        | 2.467        | 1.024        |
| EBM         | 3.100        | 1.114        |
| Hi-NAM      | 3.567        | 1.740        |
| GAM         | 4.800        | 1.030        |
| NAM         | 6.000        | 0.796        |
| Linear      | 6.367        | 1.040        |


---

Thank you once again for your valuable feedback. We hope our responses address your concerns and underscore the significance of our contributions. W have answered your individual comments in the dedicated sections below.

---


[1] Fischer, Sebastian, et al. "A curated tabular regression benchmarking suite."

---

### Author Response · Authors · 2024-12-04
**Final Remarks and Summary**

Dear Area Chair and Reviewers,

We sincerely thank the reviewers for their time and effort in evaluating our manuscript. We especially appreciate the constructive feedback and engagement provided by Reviewer ``2nBF`` during the rebuttal phase.
We are also confident that our detailed responses, references to relevant literature, and the accompanying codebase effectively address all concerns raised by Reviewer ``hMeM``. The provided evidence, theoretical grounding, and reproducible codebase robustly support the presented results.

Furthermore, the points raised by Reviewer ``FVxa`` have been carefully addressed in our revised manuscript. This includes the addition of a standalone literature review and a detailed explanation of marginal feature effects. These enhancements aim to clarify and contextualize the relevance of our contributions for readers less familiar with classical statistical concepts. Additionally, we have demonstrated NAMformer’s applicability to multi-class classification and its performance comparable to FT-Transformer.

In summary, during our rebuttal we have:
- Nearly doubled the number of datasets in our benchmarks, with results already included in the revised version showing that
    - NAMformer outperforms all other interpretable comparison models,
- Added a dedicated literature review, which will be integrated into the main manuscript in the final version,
- Provided a comparison to Integrated Gradients,
- Refined the methodology section, thanks to the insightful comments from Reviewer ``2nBF``.

---
Once again, we greatly appreciate the reviewers’ feedback. We are confident that we have thoroughly addressed the concerns and questions raised and are grateful for the opportunity to significantly improve our manuscript.

---

### Meta-Review · Area_Chair_qtx4 · 2024-12-19

**Metareview:**

This paper proposes modifications to FT-Transformers to enhance interpretability while maintaining prediction performance. The proposed approach highlights the marginal effect of each feature without performance degradation. Additionally, the analysis demonstrates that uncontextualized embeddings can effectively represent the original feature space.

The paper received mixed scores. Its strengths include the high interpretability of marginal feature networks (e.g., through simple plots of output versus input) and the diminished role of the less interpretable transformer component. The additive structure allows for a direct measurement of the transformer's relative influence compared to the marginal networks. However, several weaknesses were highlighted by reviewers:
1. The Related Work section is missing from the main paper, which hampers readability and understanding.
2. The interpretability is limited to feature-level contributions, which can be achieved by alternative methods like attention maps.
3. The role of decision trees and their influence in the encoder (e.g., thresholds in the PLE) is unclear and insufficiently discussed.
4. The evaluation is conducted on a limited number of datasets, which raises concerns about the robustness of the findings.
5. Theoretical analysis is limited, which undermines the reliability and scientific depth of the proposed approach.

The final decision is reject based on the following primary reasons: A lack of detailed analysis explaining why the proposed model provides good interpretability, such as comparisons to similar architectures. Insufficient evaluation on diverse datasets, which impacts the generalizability of the results.

Interpretable methods for tabular models are an important research direction, and we encourage the authors to address these concerns in future revisions to strengthen their work.

**Additional Comments On Reviewer Discussion:**

The paper received mixed scores: one strong positive (8) and two negatives (5, 3). The incomplete review from Reviewer 2Ug9 was excluded from the final decision.

Reviewer 2nBF found the idea of marginal feature networks valuable and maintained a positive score.

Reviewers FVxa and hMeM raised concerns about the methodology and presentation. Especially, reviewer hMeM's has concerns about the use of decision trees in the encoder, specifically their role in determining thresholds (e.g., PLE), which lacks clarity. The efficacy of the proposed simple model requires further analysis.

Both reviewers expressed concerns about the limited number of datasets used and the overall presentation quality, which are critical for establishing the robustness of a tabular model.

AC agrees with Reviewer 2nBF on the potential significance of this work for interpretable tabular models. However, detailed analysis and comprehensive evaluations are essential for a solid contribution in this area. Based on these considerations, the final decision is reject.

---

### Decision · Program_Chairs · 2025-01-22

Reject